



# The Impact of Aerosols on the Stratiform Clouds over southern West Africa: A Large-Eddy Simulation Study

Lambert Delbeke[1], Chien Wang[1], Pierre Tulet[1], Cyrielle Denjean[2], Maurin Zouzoua[3], Nicolas Maury[4], and Adrien Deroubaix[5]

[1]Laboratoire d'Aérologie, Université Paul Sabatier, CNRS, Toulouse, France
[2]CNRM, Météo-France-CNRS, Toulouse, France
[3]Laboratoire Atmosphères, Milieux, Observations Spatiales, Institut Pierre-Simon Laplace, CNRS, Guyancourt, France
[4]CNRM, Université de Toulouse-MétéoFrance-CNRS, Toulouse, now at LMD/IPSL, Paris, France
[5]Max Planck Institute, Germany

**Correspondence:** Chien Wang (chien.wang@aero.obs-mip.fr), Cyrielle Denjean (cyrielle.denjean@meteo.fr)

**Abstract.** Low level stratiform clouds (LLSCs), covering a large area, appear frequently during the wet monsoon season in southern West Africa. This region is also the place where different types of aerosols coexist, including biomass burning aerosols coming from Central and South Africa and anthropogenic aerosols emitted by local activities. We investigate the semi-direct and indirect effects of these aerosols on the diurnal cycle of LLSCs by constructing a case study based on aircraft-based

and ground-based observations from the Dynamic-Aerosol-Chemistry-Cloud-Interaction in West Africa (DACCIWA) field campaign. This case is modelled using a Large Eddy Simulation (LES) model with fine scale resolution and in-situ aerosol measurements including size distribution and chemical composition. The model has successfully reproduced the observed life cycle of the LLSC, from stratus formation to stabilization during the night, to upward development after sunrise until break-up of cloud deck in afternoon. Various sensitivity simulations using different measured aerosol profiles based on measurements

also suggest that aerosols can affect the cloud life cycle through both the indirect and semi-direct effect. Despite precipitation produced by the modeled cloud is nearly negligible, clouds lifetime is still sensitive to the aerosol concentration. As expected, modeled cloud microphysical features including cloud droplet number concentration, mean radius, and thus cloud reflectivity are all controlled by aerosol concentration. However, it is found that the difference in cloud reflectivity is not always the only factor in determining the variation of the incoming solar radiation at ground and cloud life cycle specifically beyond sunrise.

Instead, the difference in cloud-void space brought by dry air entrainment from above and thus the speed of consequent evaporation - also influenced by aerosol concentration, is an another important factor to consider. Results have shown that clouds in the case with lower aerosol concentration and larger droplet size appear to be less affected by entrainment and convection. In addition, we found that an excessive atmospheric heating up to $12\,\mathrm{K\,day^{-1}}$ produced by absorbing black carbon aerosols (BC) in our modeled cases can also affect the life cycle of modeled clouds. Such a heating is found to lower the height

of cloud top and stabilize the cloud layer, resulting a less extent in vertical development and accelerating clouds break-up. The semi-direct effect impacts on indirect effect by reducing cloud reflectivity particularly in case of polluted environment. Finally, semi-direct effect is found to contribute positively to the indirect radiative forcing due to a decreased cloud-void space, and





negatively by causing thinner clouds that would break-up faster in late afternoon, all depending on the phase in stratiform cloud diurnal cycle.

## 1 Introduction

Low-level stratiform clouds (LLSCs) are important to Earth's radiative budget through the reflection of solar radiation due to their high albedo (Hartmann et al., 1992; Chen et al., 2000) and large cloud deck covering Earth's surface more than any other cloud type (Eastman and Warren, 2014). LLSCs often occupy the upper few hundred meters in the planetary boundary layer (PBL), and their persistent appearance relies on a stable PBL that is normally associated with a large-scale subsidence above PBL under a high pressure system. LLSCs are often formed over cooler subtropical and mid-latitude oceans, constantly covering more than 50% of these areas (Wood, 2012). During the West African monsoon season, LLSCs frequently form over continental southern West Africa (SWA) in the night and would likely break up in the early afternoon of the following day there (Schrage and Fink, 2012; Schuster et al., 2013). LLSCs are, under polluted conditions characterized by numerous and small cloud droplets, increasing the cloud albedo, suppressing drizzle, and extending lifetime (Twomey, 1957; Haywood and Boucher, 2000; Liu et al., 2014; Carslaw et al., 2017). The presence of LLSCs impacts the radiative budget of atmospheric boundary layer and surface and also affect the diurnal cycle of the convective boundary layer and thus the regional climate (Knippertz et al., 2011; Hannak et al., 2017). However, the diurnal cycle of LLSCs is still poorly represented in weather and climate models, especially over SWA, because the processes behind the variability of LLSCs cover remain elusive (Knippertz et al., 2011; Hannak et al., 2017; Hill et al., 2018).

Stratiform clouds are sensitive to aerosol properties (concentration, chemistry) and vertical distribution. Aerosols inside stratiform clouds are also modified by physico-chemical processes which can influence the aerosol concentration (Wood, 2012). Interactions between aerosols and clouds, and their effects on radiation, precipitation, and regional circulations, remain one of the largest uncertainties in understanding climate. Indeed, the indirect effect forcing is still difficult to estimate (IPCC 2021) and climate model struggle to minimize uncertainties (Li et al., 2022). Some aerosol particles as black carbon absorb a substantial amount of shortwave radiation, which results in rapid atmospheric thermodynamic adjustments. This semi-direct aerosol radiative effect can be positive or negative depending on the relative distribution of the aerosol with respect to clouds. Several previous studies were conducted to investigate aerosol-clouds interactions of LLSCs using high-resolution LES models but mainly over ocean (Ackerman et al., 2004; Sandu et al., 2008; Twohy et al., 2013; Flossmann and Wobrock, 2019) where surface fluxes are having little diurnal variation, moisture is mainly provided by evaporation from sea surface to maintain the stratiform cloud layer. By contrast over land, moisture supply is dependent on the characteristics of the surface (Wood, 2012).

During the West Africa Monsoon (WAM), aerosols can come from both local and remote sources to SWA. Large amount of Biomass Burning Aerosols (BBA) are transported from southern and Central African towards SWA during the summer monsoon (Haslett et al., 2019). These air masses are further loaded with additional aerosols from anthropogenic emissions





upon reaching the highly urbanized regions near the coast. (Chatfield et al., 1998; Sauvage et al., 2005; Mari et al., 2008; Murphy et al., 2010; Reeves et al., 2010; Menut et al., 2018; Haslett et al., 2019). A significant quantity of wind-blown mineral dust aerosols emitted from the Sahara and Sahel throughout the year with a peak in springtime (Marticorena and Bergametti, 1996) can also reach SWA far south often in June (Knippertz et al., 2017). Local sources of aerosols in SWA are related to

anthropogenic activities near the coast from where polluted plumes transport inland (Deroubaix et al., 2019). These emissions are supposed to increase with the expected growing of the population (Liousse et al., 2014). These different sources of aerosols give a complex mix of species with a high loading, having a serious impact on human health (Bauer et al., 2019) and also on the diurnal cycle of winds, LLSCs as well as precipitation over SWA (Taylor et al., 2019).

The DACCIWA project was designed to better characterize cloud-aerosol-precipitation interactions in SWA (Knippertz et al., 2015). The measurement campaign provides a comprehensive set of ground-based and airborne-based measurements of clouds and aerosols in June-July 2016. (Knippertz et al., 2017; Kalthoff et al., 2018; Flamant et al., 2018) and model analyses. Measurements were conducted at three supersites, Savè (Benin), Kumasi (Ghana) and Ile-Ife(Nigeria) (Fig.1), and coordinated with three research aircrafts : the French ATR-42 operated by SAFIRE (Service des Avions Français Instrumentés pour la

Recherche en Environnement), the British Twin Otter operated by British Antarctic Survey and the German Falcon aircraft operated by DLR (Deutsches Zentrum für Luft und Raumfahrt). Additional radiosoundings were launched from Savè with high temporal frequency, which specifically benefits the monitoring of the LLSCs diurnal evolution.

Taylor et al. (2019) and Denjean et al. (2020a) showed that the majority of cloud condensation nuclei and absorbing aerosol

were from ubiquitous long-range transported BBA, causing a polluted background which limits the effect of local pollution on cloud properties and aerosol radiative effects. Modeling studies have suggested that light-absorbing aerosols from combustion sources like BBA and anthropogenic sources can impact the formation, evolution and precipitation of LLSCs, especially over south-eastern Atlantic. Using COSMO-ART model in a simulation for 2-3 July 2016 case, Deetz et al. (2018) found that under the influence of the Maritime Inflow (MI, cold air) from Guinean Gulf, stratus-stratocumulus transition are susceptible to the

aerosol direct effect resulting in a spatial shift in the MI front and a temporal shift of the cloud transition. Over SWA and influenced by anthropogenic emission sources, the breakup time of LLSCs can be delayed by one hour and daily precipitation rate can decrease by 7.5% according to Deroubaix et al. (2022). Moreover, semi-direct and indirect effects were studied together by varying respective magnitude or forcing emission of anthropogenic aerosols but they were not examined separately. Haslett et al. (2019) denote that cloud droplets number concentration increases up to 27 % due to transported BBA using COSMO-

ART and make cloud and rain less sensitive to future increase in anthropogenic emissions on regional scale. The impact of sedimentation on LLSCs has been studied by Dearden et al. (2018) using the Met Office NERC Cloud model (MONC) who highlight that cloud droplets size has an effect on liquid water path. Menut et al. (2019) showed with WFR-CHIMERE that a decrease of anthropogenic emission along the SWA coast lead to a northward shift of the monsoonal precipitation and the increase of surface wind speed over arid region in the Sahel resulting in a increase of mineral dust emission. These previous

modeling studies all highlight in a regional scale and take care in majority of aerosol chemical composition but they do not



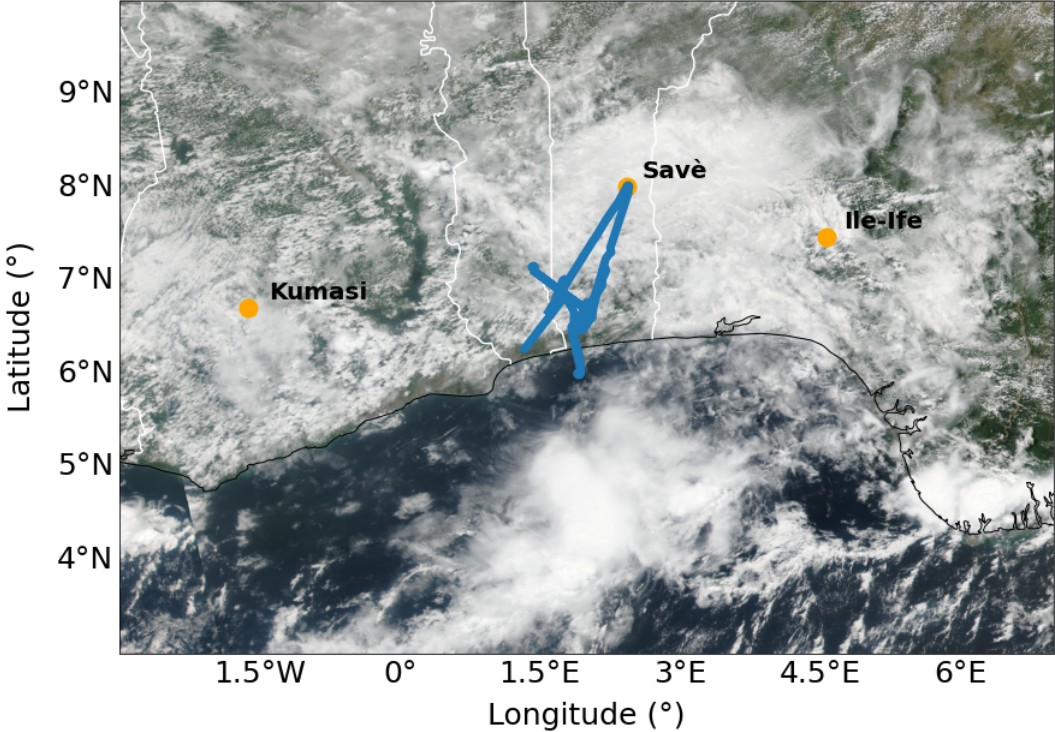

**Figure 1.** Map of southern West Africa with Savè, Kumasi and Ile-Ife locations and the flight track (blue line) of the ATR-42 the 3 July 2016 with NASA Suomi NPP/VIIRS true color corrected reflectance (https://worldview.earthdata.nasa.gov/).

take into account all aerosol species detected during the campaign especially BC. Pedruzo-Bagazgoitia et al. (2020) analyzed the stratocumulus-cumulus transition at fine scale (a dozen of kilometer sidelong) using a LES at high resolution (50x50 m$^2$) but without considering aerosols effects.

The aim of this study is to understand the relative impacts of local and transported aerosols on the life cycle of LLSCs during the monsoon period over SWA by using observational data obtained from the well-documented DACCIWA field campaign alongside a high-resolution LES model including interactive aerosol module that is able to represent the complex aerosol composition. This modelling case is focused on local scale and fine resolution over SWA which made it one of the few studies that model and analysed stratiform cloud diurnal cycle over land when a majority take place over ocean. For this purpose,
using observational data we firstly identified a reference case for modeling, that is a LLSCs case observed on July 3, 2016 at Savè site. The short description of observations, data, and the model as well as configurations of different simulations will be presented in the Method section after the Introduction. Then an analysis will be driven to understand and validate the reference case compared to measurements. Then, to assess the aerosol effects on stratiform clouds and based on observed aerosol data, we constructed several different aerosol profiles which differ in term of aerosol size distribution and chemistry





for sensitivity modeling studies. A first sensitivity analysis will be driven to asses the impact of aerosol concentration and consequently the indirect effect. Another analysis will be focused on the impact of aerosol optical properties by switching off aerosol (semi) direct contributions to radiative budget to exhibit the relative changes imposed by direct and semi-direct effects. In the DACCIWA framework, such analysis is a first and differs from other modelling studies by performing this set of scenarios and configurations in order to better investigate on indirect and semi-direct effects of aerosols from biomass burning and local anthropogenic sources. Finally, this study will conclude by a summary of findings.

## 2 Methods

### 2.1 Observations

The relevant measurements of the DACCIWA field campaign used to select our LLSC case and to configure the model simulations are described as follows:

i) Radiosondes were launched with the MODEM system every 1 to 1.5 hour between 17:00 and 11:00 UTC at the supersite of Savè in Benin. It is located at 185 km from the coast and 166 m above mean sea level (a.m.s.l) where the area is rather flat, and the vegetation is mainly composed of small trees and shrubs. Temperature, pressure, relative humidity and wind vertical profiles in the lower atmosphere reaching a maximum height of around 1500 m a.g.l (above ground level) were measured with a 1 s temporal resolution (4-5m of vertical resolution) (Derrien et al., 2016). These sondes use two ballons of different volumes to reach a preset time of ascent and after the cutting of the larger balloon, the second allows to retrieve the sonde for another use (Legain et al., 2013).

ii) At the supersite of Savè, meteorological parameters were measured using different instruments. A CHM15k Ceilometer was deployed by the Karlsruher Institut für Technologie (KIT) to measure the cloud base height continuously with a 1 min resolution and a 15 m vertical resolution. Three cloud base heights are recorded from the backscatter profiles produced by the lidar with a wavelength of 1064 nm and a 5-7 kHz rate (Handwerker et al., 2016). The cloud cover was monitored every day by using a MOBOTIX S15 cloud camera, installed by UPS (Université Paul Sabatier) team, to obtain pictures in visible and IR every 2 min. The aperture angles for the IR channel corresponds to a 158 m x 114 m area at a height of 200 m. Pictures are coded in RGB components over 256 colors. A pixel is colored depending on the emissivity of the sky and so its brightness temperature. So a low cloud base is seen as red and a clear sky is seen as blue. A microwave radiometer (absolute humidity and air temperature profiler HATPRO-G4 from Radiometer Physics GmbH) was installed by KIT to measure brightness temperature to retrieve absolute humidity, liquid water path, and air temperature. The surface fluxes and net radiations were measured with an energy balance station deployed over grass and bushes. Additional measurements were also soil heat flux, air density and turbulence parameters as well as sensible and latent heat flux.





iii) The aircraft campaign took place from 29 June to 16 July 2016. It was a collaborative work between three research aircrafts but in this study only the ATR-42 is selected as it flew around Savè between 10:00 and 11:00 UTC and probed the cloud layer. The cloud droplet size distribution was measured with a cloud droplet probe (CDP) (Taylor et al., 2019). The chemical composition for non-refractive compounds was measured with an Aerodyne compact Time-of-Flight Aerosol Mass Spectrometer (HR-ToF-AMS) analyzed (Brito et al., 2018). BC mass concentration was measured with a single particle soot photometer (SP2) (Denjean et al., 2020b). The aerosol number size distribution was measured with a custom-built scanning mobility sizer spectrometer (SMPS, 20–485 nm), an ultra-high sensitivity aerosol spectrometer (UHSAS, 0.04–1 $\mu$m), and an optical particle counter (OPC GRIMM model 1.109, 0.3–32 $\mu$m) corrected for the complex refractive index provided in Denjean et al. (2020a). The total concentration number of particles larger than 10 nm was measured by condensation particle counter (CPC, model MARIE). A suite of instruments measured also meteorological variables such as temperature, humidity, pressure and winds. Gas concentration analyzer measured some gas like CO2, CH4, CO.

## 2.2 Description of the studied case

Our study analyzes the diurnal cycle of LLSCs based on the case study of 3 July 2016 at the Savè supersite. The cloud deck formed during the night, at around 02:00 UTC, close to the core of the Nocturnal-Low-Level Jet (NLLJ) where the cooling is maximum (Lohou et al., 2020) and could reach a maximum speed of around $6 \mathrm{~m} \mathrm{~s}^{-1}$ (Kalthoff et al., 2018). At formation, the cloud base and top heights are located around $310 \pm 30 \mathrm{~m}$ and $640 \pm 100 \mathrm{~m}$, respectively, and were maintained due to the cloud top radiative cooling and cold advection (Dione et al., 2019).

The diurnal cycle of LLSCs over SWA typically involves four phases: stable phase, jet phase, stratus phase, and convective phase (Dione et al., 2019; Lohou et al., 2020). The stable phase begins just after sunset and is characterized by a weak monsoon flow and the cessation of buoyancy-driven turbulence (generated by surface heating) within the atmospheric boundary layer (ABL) (Zouzoua et al., 2021). The jet phase corresponds to the settlement of key drivers of cooler air advection. Maritime inflow (MI), cold and slightly humid air from the Guinean coast reaches Savè at the end of the afternoon (between 16:00 UTC and 20:00 UTC) then the NLLJ formation occurs (Adler et al., 2019). The stratus phase begins with LLSC formation when the advective cooling continuously increases the relative humidity (RH) until saturation is reached between 22:00 and 06:00 UTC. Turbulent mixing maintains cooling below the cloud base and strong radiative cooling at the cloud top leads to the persistence of a thick stratus layer (Schuster et al., 2013; Babić et al., 2019). The LLSCs life cycle ends during the final convective phase which begins when the ABL develops vertically due to solar heating at the surface. By using dataset from Savè supersite Zouzoua et al. (2021) identify three scenarios of evolution depending on the LLSCs coupling to the surface at sunrise. The coupling was assessed by the departure between the Cloud Base Height (CBH) and the Lifting Condensation Level (LCL).

The LLSCs observed on 3 July 2016 follow the four aforementioned phases and evolves by scenario C described by Zouzoua et al. (2021) as seen in Figure 2; the cloud is coupled to the surface at sunrise (06:30 UTC) and its base rises with growing ABL until its break-up in the late afternoon (around 16:00 UTC). This cloud deck stands longer (2-3 hours more) compared to



other LLSCs observed during the campaign. Co-located radar at Savè supersite allows the detection of light precipitation from higher clouds during the first hours of the convective phase while no precipitation was detected by the surface rain gauge. Thus, this late LLSC break-up could be explained by a significant increase in its liquid water content (LWC) caused by evaporation of this light precipitation (Zouzoua et al., 2021).

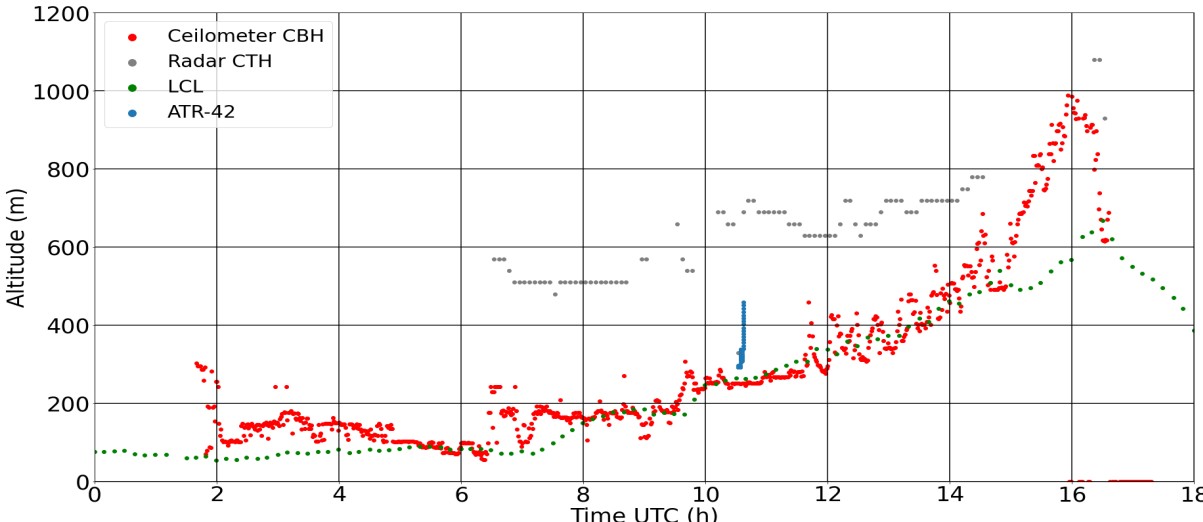

**Figure 2.** 3 July cloud evolution with the representation of the Cloud Base Height (CBH), the Cloud Top Height (CTH), LCL and ATR-42 flight track near Savè.

On 3 July 2016, the ATR-42 flew around Savè supersite and probed the boundary layer around 10:00 UTC. The measurements confirmed an important aerosol number concentration with a maximum of around $3500\ \mathrm{cm}^{-3}$ mainly located in the Aikten mode. The ATR-42 detected an export of pollution from Lomé (a coast city) which could explain the higher loading of aerosol in the Aikten mode (Denjean et al., 2020a). The aerosol chemical composition was mainly dominated by organics

(55.3%), followed by sufates (24.5%), ammoniac (11.2%) and nitrates (6.2%) while little BC mass was detected around Savè (2.8%). However, these data are directly extracted from DACCIWA database and the size distribution has to be corrected from the aerosol refractive index to avoid bias. For this purpose, Denjean et al. (2020a) provided corrected profiles for different typical aerosol population encountered during the DACCIWA campaign.

### 2.3 Meso-NH Model

In this study, we have simulated the observed case using the French model Meso-NH (Lac et al., 2018). Meso-NH is a non-hydrostatic atmospheric research model that has been applied to studies in scales, ranging from synoptic to turbulent. Employed in a limited area, the model uses advanced numerical techniques like monotonic advection schemes for scalar trasnport and fourth-order WENO advection scheme for momentum (Jiang and Shu, 1996). Sub-grid turbulence is parametrized using





turbulence kinetic energy (TKE) based on Deardorff turbulent mixing length. The advection scheme used for horizontal and
vertical velocities is CEN4TH, a 4th order advection scheme centred on space and time applied with a Runge-Kutta centred
4th order temporal scheme for momentum advection. Aerosol and chemistry are also well represented. Here, Meso-NH version 5.4.2 (www.mesonh.aero.obs-mip.fr) is used and the relevant component modules and parametrizations for this study are
described as follows:

LIMA (Liquid Ice Multiple Aerosol) is a complete two-moment scheme (Vié et al., 2016) predicting both the mass mixing
ratio and the number concentration of water species. Based on ICE3-ICE4 schemes (Caniaux et al., 1994; Pinty and Jabouille,
1998; Lascaux et al., 2006) and the two-moment warm microphysical scheme C2R2 from (Cohard and Pinty, 2000), LIMA
predicts the number concentration of cloud droplets, raindrops and pristine ice crystals. It includes a prognostic representation
of aerosol population using a superimposition of several aerosol modes with each mode defined by its chemical composition,
size distribution and aerosols can act as a Cloud Condensation Nuclei (CCN) or an Ice Freezing Nuclei (IFN). A variant to
C2R2, called KHKO, was developed by Geoffroy et al. (2008) for clouds producing drizzle following Khairoutdinov and Kogan (2000) parametrization. These clouds are low precipitating warm clouds, and not sufficiently thick to produce heavy rain.
The precipitating hydrometeors are drizzle only and their diameter are of the order of several dozens of micrometers. These
modifications for KHKO were brought inside LIMA warm phase in order to better represent drizzle completing LIMA.

ECMWF radiation module, originated from ECMWF and based on two-stream methods, calculates the atmospheric heating
rate and the net surface radiative forcing. Longwave (LW) radiation scheme used is Rapid Radiation Transfer Model (RRTM,
Mlawer et al. (1997)), based on the correlated k-distribution method. It integrates 16 bands and 140 g points (Morcrette, 2002).
The Shortwave (SW) scheme uses the photon path distribution method (Fouquart and Bonnel, 1980) in six spectral bands.
Fluxes are calculated independently in clear and cloudy portion before being aggregated. The liquid cloud effective radius is
computed from the liquid water content with the Martin et al. (1994) parametrization.

ORILAM (Organic Inorganic Lognormal Aerosols Model) is an aerosol module coupled to Meso-NH and connected to
LIMA (Tulet et al., 2005). It describes the size distribution and the chemical composition of aerosols using two lognormal
functions for the Aitken and accumulations modes. These modes are internally mixed and for each of them the model computes
the evolution of the primary species (black carbon and primary organic carbon) , three inorganic ions ($NO_3-$, $NH_4+$, $SO_4 2-$)
and condensed water. ORILAM includes Second Organic Aerosols (SOA) (Tulet et al., 2006) but are not taken into account
in this study. Three moments (zeroth, third and sixth) are considered for each mode to compute the evolution of total number,
median diameter and geometric standard deviation. The size distribution can evolve through a particle coagulation process with
both intramodal and intermodal calculations. It can also evolve through condensation and merging between modes. ORILAM
includes the CCN activation scheme of Abdul-Razzak and Ghan (2004) in order to replace the one of LIMA to calculate the
number of activated CCN. The others LIMA parametrizations in warm phase like the calculation of drizzle remain active. The
use of ORILAM needs to activate the gas phase chemistry scheme of Meso-NH (Tulet et al., 2003; Mari et al., 2004) using





the EXQSSA solver. Inorganic chemistry system (EQSAM, Metzger et al. (2002)) solves the chemical composition of sulfate-nitrate-water-ammonium aerosols based on thermodynamics equilibrium. For secondary organic aerosols, the thermodynamic equilibrium uses the MPMPO scheme from Griffin et al. (2003). ORILAM computes directly the evolution of aerosol extinction, SSA and asymmetry factors that are coupled online with the radiation scheme of Meso-NH for the 6 short wavelengths from the aerosol chemical composition and size parameters (Aouizerats et al., 2010).

SURFEX is a standardized surface module containing surface schemes externalized of Meso-NH (Masson et al., 2013). Each grid can be split into four tiles: land, town, sea and inland water (lake, rivers). In case of a shrubs typical surface, the interactions between soil, biological and atmosphere are calculated by ISBA parametrization (Noilhan and Planton, 1989). It represents the effect of vegetation and bare soil. Several evapotranspiration formulations are available for plants and for simulating the CO2 fluxes. Soil is represented as a bucket of two or three layer. The land tile can be separated up to 19 subtiles following the type of vegetation.

## 2.4 Model settings

Based on observations and the capability of the model, a reference case (REF) was first designed to simulate through LES. The reference case serves as a base to reproduce the major features of observed LLSCs diurnal cycle particularly under an observed aerosol profile. It also serves as a reference for further sensitivity simulations with different aerosol configurations to study the impacts of aerosol composition alongside abundance on LLSCs.

The domain is a 3D box of 9.6 km x 9.6 km x 2 km in size with a horizontal resolution of 40 m x 40 m. The vertical resolution is 10 m between 0 m and 1200 m then 40 m above until 2 km of altitude. Such high resolution is able to resolve explicitly the biggest turbulent eddies. A periodic boundary condition on the horizontal directions is applied and an absorbing layer is set at 1.8 km. A thermodynamic perturbation is deployed to activate turbulence at the beginning of the simulation at 23:00 UTC of 2 July and the spin-up is 1h. A subsidence profile is applied following Bellon and Stevens (2013) scheme $w_{subs}(z) = -w_0(1 - e^{\frac{-z}{z_w}})$, with $w_0 = 15 \ \text{mm s}^{-1}$ and $z_w = 250 \ \text{m}$. This subsidence profile is applied during the entire simulation to keep a nearly constant cloud top height during the stratus phase and to better control the convective phase. The surface energy and water fluxes are simulated by SURFEX ISBA scheme parametrized by data from Savè supersite measurements with the typical vegetation around of Savè which consists of shrubs, crops or taller trees assuming a flat surface corresponding to the area around Savè. A time-step of 2 s is used which appears to be adequate to study accurately the LLSCs diurnal variations. Note that the radiation scheme is called every 10 minutes.

REF case is configured using the radiosounding profile of 2 July at 23:00 UTC for temperature, humidity and horizontal wind components (U,V). The simulation is then controlled by tendency profiles of temperature and humidity applied homogeneously on the domain each hour. These tendency profiles are based on the radiosoundings profiles launched on 3 July between 00:00





and 11:00 UTC. After 11:00 UTC, the next tendency profiles were designed based on the measurements of the microwave radiometer, the analysis of resulted surface incoming solar radiative flux, cloud thickness and cover.

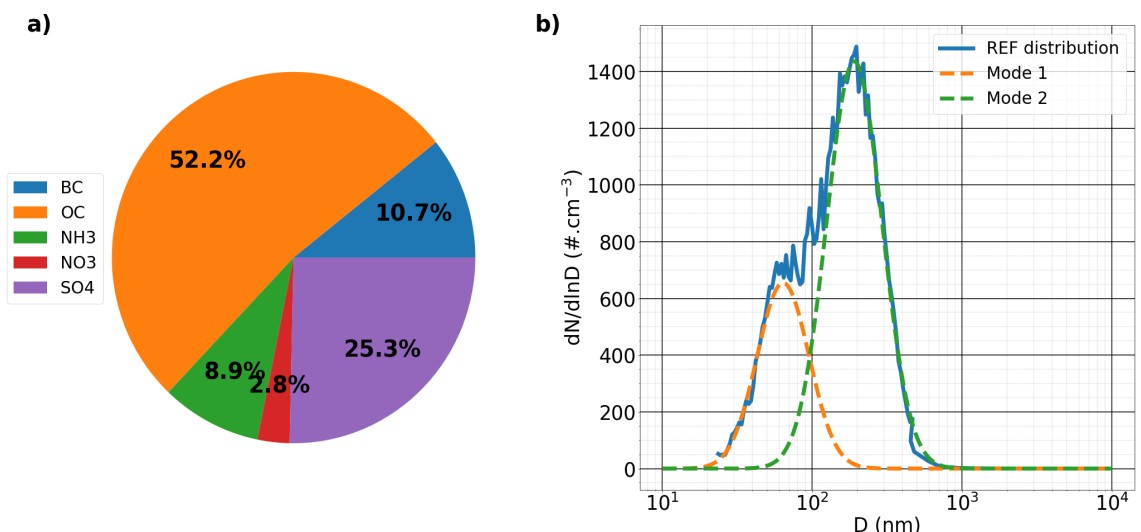

**Figure 3.** REF mass composition (a) and size distribution fitted into 2 modes described in Table 1 (b).

|  | $N_a$ (cm$^{-3}$) | $\sigma$ | D (nm) |
|---|---|---|---|
| Mode 1 | 654 | 1.49 | 63.98 |
| Mode 2 | 1530 | 1.53 | 190.97 |

**Table 1.** REF aerosol size distribution describes by two modes configured by three parameters (number concentration, standard deviation and diameter).

We decided to use a "background" distribution as the aerosol profile for REF simulation. This profile, described in Denjean et al. (2020a), actually reflects the influence of aged BBA on clouds with minor influence of local anthropogenic sources. The aerosol number size distribution is dominated by a particle accumulation mode centered at 190 nm and a smaller Aiken mode centered at 64 nm as seen in Figure 3b. This profile exhibits a high loading of aerosols with a maximum of 1400 cm$^{-3}$ detected in the accumulation mode. The aerosol chemical composition was dominated by organics (52.2%) followed by sulfates (25.3%), ammonium (8.9%), BC (10.7%) and nitrate (2.8%). The configuration of ORILAM have been initialized using the REF aerosol chemical composition and number size distribution given in Table 1 and Figure 3b by fitting the SMPS profiles into two lognormal modes using the "py-smps" package (Hagan et al., 2022) with each mode having the same chemical composition.





## 3 Analysis of REF Results

### 3.1 Simulated LLSCs evolution

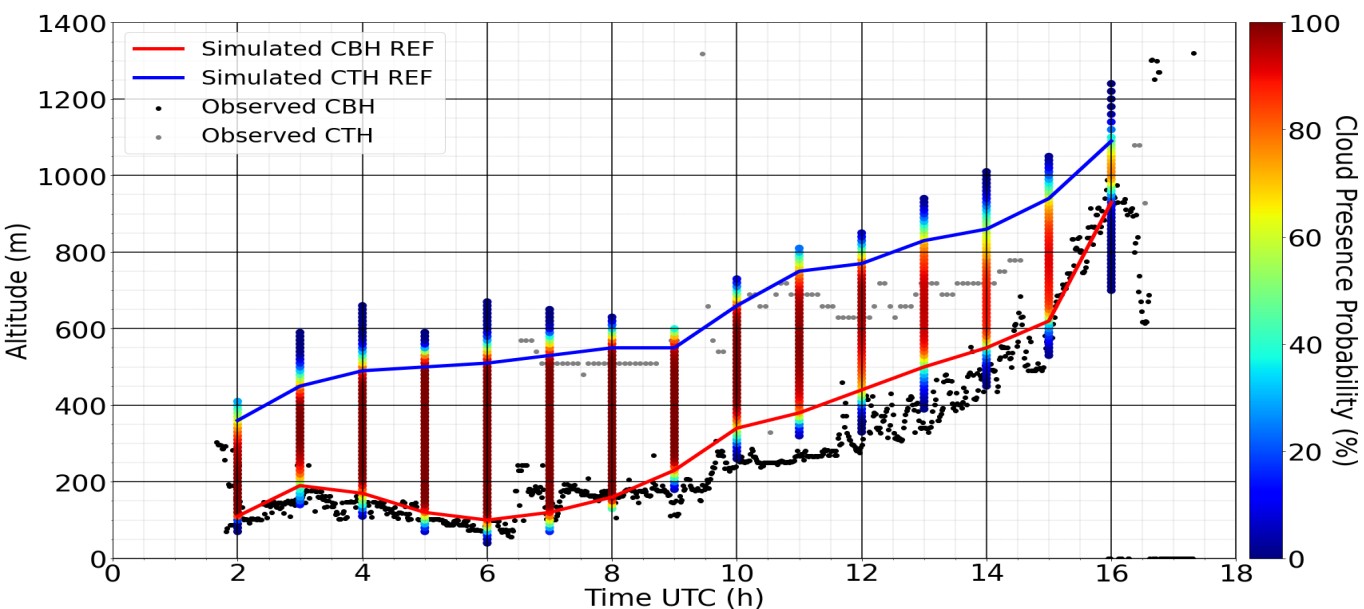

**Figure 4.** Simulated LLSCs deck evolution compared to Savè ceilometer and radar measurements, vertical colored bars attribute at each altitude level a modeled cloud presence probability. Here mean CBH and CTH represents cloud base and cloud top height, respectively.

The simulation of the REF scenario reproduces the formation of observed LLSCs deck on 3 July 2016 as shown in Figure 4. The formation of clouds leads, as described in section 2.2, to the end of the jet phase. The mean CBH estimated from the mixing ratio of cloud droplets follows the ceilometer's measurements during the stratus phase between 2:00 and 10:00 UTC, varying between 100 and 200 m of altitude. The simulated mean Cloud Top Height (CTH) evolves from 400 to 550 m of altitude, in range of the values from 500 to 580 m detected by the radar. During the convective phase, the model results differ slightly from the observations.

Nevertheless, the mean simulated CBH and CTH are overestimated compared to ceilometer and radar values in some period particularly late morning and afternoon. The simulated CBH can differ from ceilometer one by 150 m of altitude at 11:00 UTC. The CTH is often overestimated by 100 m. Between 15:00 and 16:00 UTC, the mean CBH follows again the detected values (600 to 950 m) (no radar values are available to validate the simulated CTH). As mentioned in section 2.1, the ceilometer is a lidar while the radar values are derived from reflectivity vertical profiles which have a 30 m of resolution. The differences between the model and the observation between 13:00 and 16:00 UTC are likely due to the domain averaging of the simulated values, the tendency profiles established from corrected radiosonded values, the ceilometer values limited to only one vertical direction, or the limitation of radar in detecting hydrometeors.





**Figure 5.** Comparison between modeled liquid water path (LWP, g kg$^{-1}$ m) and the images from Savè cloud camera at 06:00 (top), 12:00 (middle) and 16:00 UTC (bottom).

To analyze the cloud cover over the domain, Cloud Presence Probability (CPP) at each altitude level is calculated by taking

all pixels which exceed 0.05 g kg$^{-1}$ in total condensed water mixing ratio as a cloud pixel at each altitude level. Liquid water path (LWP) at each column calculated based on cloud pixels brings a view on the horizontal organization and homogeneity of the cloud deck. During the stratus phase, the CPP is near equal to 100% between CBH and CTH giving a homogeneous cloud deck. The top panel of Figure 5 gives a comparison of the cloud organization in the model and from cloud observation with visible camera.





At 6:00 UTC, it can be seen that cloud deck covers the entire domain in both model and observation (note the clear cloud rolls in model results). Between 10:00 and 13:00 UTC, the CPP at the mid distance from mean CBH and CTH decreases from near 100% to 90%. Near the two averaged values, CPP decreases more to reach near 60% and 80% at CBH and CTH, respectively. This leads to a more inhomogeneous cloud deck confirmed by the LWP map and the observation of the cloud camera at 12:00 UTC shown in middle row of Figure 5. Indeed, more no-cloud pixels begin to appear between clouds and the sunlight is seen through the cloud deck by the camera. Finally, the CPP continue to decrease until the end of convection phase by reaching a maximum of near 80% and around mean CBH and CTH it reaches 20% and 40% respectively. This demonstrates the break-up of cloud deck during convection and the decreasing of cloud thickness. The bottom panels of Figure 5 show clearly the break-up of clouds at 16:00 UTC. The LWP map shows numerous thin clouds corresponding to those seen by the camera of Savè.

Figure 6a shows the comparison between the SW radiation flux at surface (SWRADSURF) averaged over the modeled domain and measurements performed by the energy balance station. Observed values are fitted following the locally Weighted Scatterplot Smoothing (LOWESS) method (Cleveland, 1979), which is a non-parametric regression method performing weighted local linear fits. The temporal evolution of the modeled SWRADSURF follows well the observations although some biases can be observed. After 06:00 UTC the solar radiation reaches the ground and as the cloud deck thickness and covering show little variations, the radiative flux increases gradually by reaching near $200 \text{ W m}^{-2}$ at the end of the stratus phase (10:00 UTC). As the clouds deck becomes inhomogeneous during the convective phase (10:00 to 16:00 UTC), the solar flux reaches a maximum of $300 \text{ W m}^{-2}$, which is a bit less than the fitted $350 \text{ W m}^{-2}$ value. Finally, when the clouds break-up, more solar radiation can reach the surface. After this period during which model and observation agree well, from 15:00 UTC the mean curve decreases to $200 \text{ W m}^{-2}$ while the fit curve is near $320 \text{ W m}^{-2}$ due to an overestimation of the cloud thickness by the model. At 16:00 UTC, both modeled and measurement values are very close around $280 \text{ W m}^{-2}$. But, as the LLSCs breakup later than common clouds during DACCIWA campaign (more than 2-3 hours), it implies that modeled SWRADSURF is low. Generally, the modeled maximum values are higher than the ones detected by Savè instrument. For example, at 10:00 UTC, the balance detected a peak of $300 \text{ W m}^{-2}$ while the model value reached near $400 \text{ W m}^{-2}$. Such difference is reduced during the convective phase.

Figure 6b and 6c shows that the evolution of modeled latent and sensible heat fluxes well reproduced those measured by the instrument as shown in the middle and bottom row respectively of Figure 6. During the night, the sensible heat flux was negative then increased to $0 \text{ W m}^{-2}$ close to the sunrise time (6:00 UTC), indicating a reduction of the cooling close to the ground (Dione et al., 2019). Between 9:00 and 14:00 UTC, the modeled two heat fluxes followed the measured trends though overestimated by almost 70 and $18 \text{ W m}^{-2}$. Then the mean modeled curves go below the fitted observed curves at 15:00 UTC and finally decrease to almost $0 \text{ W m}^{-2}$ after 18:00 UTC. The difference between modeled and observed latent and sensible heat fluxes may be due to the different area covered by the measurements and the model and the prescribed subgrid-scale distributions of cloud droplets.




**Figure 6.** Comparison between Savè surface observation and REF simulation for SW radiation flux at surface (SWRADSURF,a), sensible heat flux (H,b) and latent heat flux (LE,c) all expressed in W m$^{-2}$ at the surface. The variation of REF for each parameter indicates the range of possible values these parameters can take.

In summary, the REF simulation has successfully reproduced the major observations obtained by the instruments at Savè on 3 July 2016. The modeled cloud thickness and coverage represent well the cloud situation with some inaccuracies due to the lack of data and insufficient correction of the tendency profiles applied all along the simulation to control temperature and humidity every hour. The modeled heating of the ground by solar radiation also follows the measurements of the energy



balance of Savè and maximum variation on the domain are a bit overestimated. The sensible and latent heat flux detected at Savè have also been well captured by the model.

## 3.2 Thermodynamic, dynamical, cloud, microphysic and radiative analysis

Thermodynamic, dynamical, and radiative processes and their interaction with cloud microphysic are among the key factors in determining the life cycle of LLSCs. Here we discuss the evolutions of these processes simulated by the model in the REF case in order to better understand the reasons behind model-observation consistency or discrepancy. The discussion will be emphasized in three periods. The first period is the transition between jet and stratus phase (between 00:00 and 04:00 UTC) to observe how clouds are formed. The second period is the stratus phase between 06:00 and 10:00 UTC which is interesting due to the cloud layer stability observed by the instruments of Savè. The third period is the convective phase between 12:00 and 17:00 UTC to study how the properties of LLSCs evolve during the break-up stage.

### 3.2.1 Transition jet-status phase

The formation of clouds is controlled by the temperature and humidity tendency profiles established from the radiosonded measurements made on the 3rd of July 2016. Figure 7 gives domain-averaged profiles of the simulated temperature (T) and relative humidity (RH). As explained in section 2.2, maritime inflow already reached the site that increased humidity and created NLLJ. Temperature decreases from 24°C to 23°C at ground from 00:00 to 04:00 UTC and from 24°C to 21°C near 400 m of altitude. The advection of cold and slightly humid air leads to the increase of RH as expected reaching 100% at 02:00 UTC at 100 m. After this time, RH exceeds saturation between 100 and 500 m of altitude. The inversion occurs around 325 m and 500 m respectively at 02:00 UTC and at 04:00 UTC. The NLLJ is well represented as the mean wind speed ($w_s$) before cloud formation is more than $7 \, \mathrm{m \, s^{-1}}$. After formation of cloud, the NLLJ core corresponds near to the mean cloud base height (Adler et al., 2019; Babić et al., 2019; Lohou et al., 2020). The turbulence during this period is shear-driven due to this NLLJ which well mix the sub-cloud layer. The TKE is high above ground (0.2 to $0.5 \, \mathrm{m^2 \, s^{-2}}$), then decreases to near zero along altitude at 00:00 UTC. After 02:00 UTC, TKE increases at the level of the CTH (350 and 500 m) and decreases at the center of clouds ($0.04 \, \mathrm{m^2 \, s^{-2}}$), indicating this area is less turbulent than the extremities of the cloud layer.

Cloud droplet number concentration ($N_c$) is determined by the supersaturation in an updraft and the concentration of aerosols that activate at their supersaturation. In Figure 7e, simulated aerosol concentrations ($N_a$) were the highest close to the ground and decreased with altitude up to around 2 km. This simulated aerosol profile is similar to those observed by airborne measurements during DACCIWA (Taylor et al., 2019; Denjean et al., 2020a) with locally emitted aerosols transported above the boundary layer due to a combination of land-sea surface temperature gradients, orography-forces circulation and the diurnal cycle of the wind along the coastline (Deroubaix et al., 2019; Flamant et al., 2018). The simulated clouds microphysic reflect polluted conditions with $N_c$ reaching $1750 \, \mathrm{droplets \, cm^{-3}}$ and $r_c$ around 5 μm which is not enough to form even drizzle (size between 0.2 mm and 0.5 mm, (Pruppacher et al., 1998; Sandu et al., 2008)). These values are in the range of those measured inland at the same altitude by Taylor et al. (2019) during DACCIWA. Median of simulated $N_c$ was $500 \, \mathrm{droplets \, cm^{-3}}$ at the




beginning of cloud formation and reached $1750\,\mathrm{droplets\,cm^{-3}}$ latter, most likely due to the continuous activation of aerosol into cloud droplets.

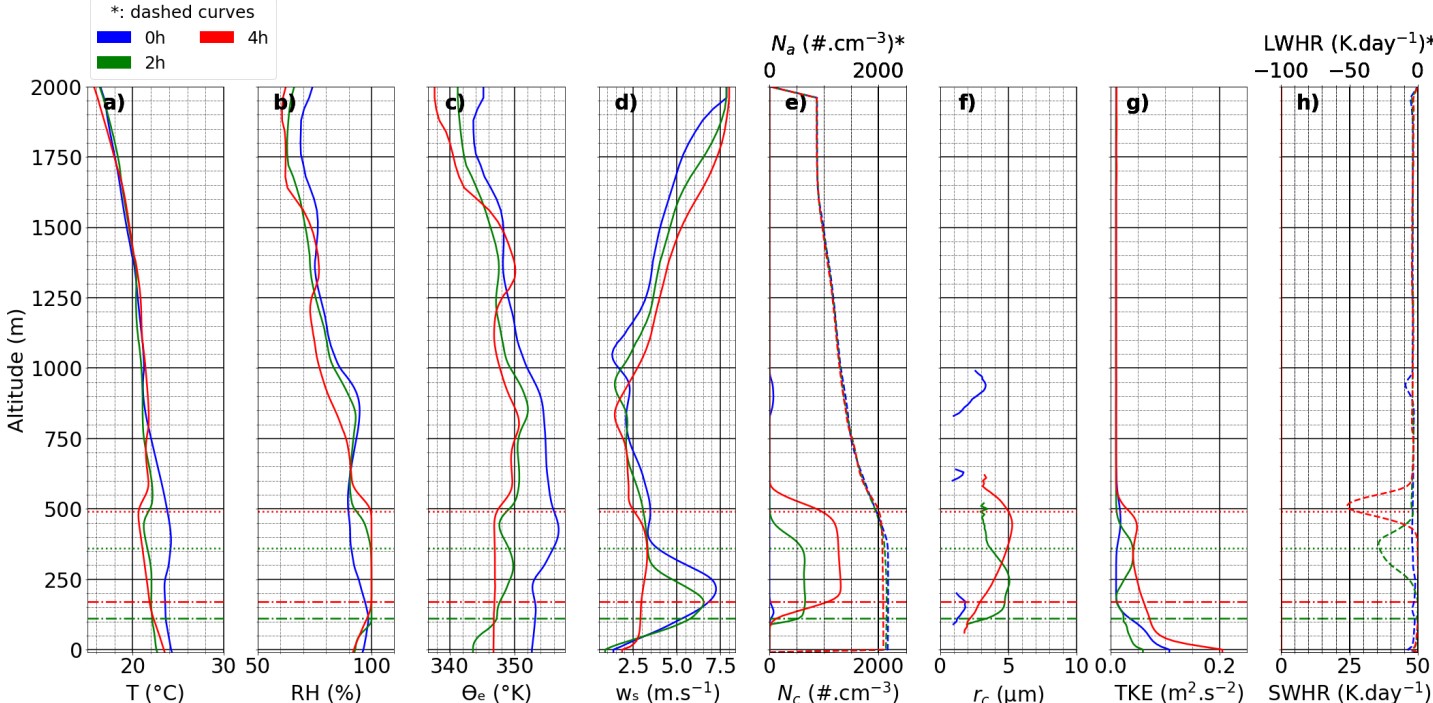

**Figure 7.** Profiles from left to right of temperature (T, a), relative humidity (RH, b), equivalent potential temperature ($\theta_e$, c), horizontal wind speed (w_s, d), aerosol number concentration ($N_a$, dashed curve, e), cloud droplets number concentration ($N_c$, plain curve, e), cloud droplet radius ($r_c$, f), turbulent kinetic energy (TKE, g), longwave heating rate (LWHR, dashed curve, h) and shortwave heating rate (SWHR, plain curve, h) at 00:00, 02:00 and 04:00 UTC. Dashdot horizontal lines represent mean cloud base height (CBH) and dotted horizontal lines the mean cloud top height (CTH).

The emission of thermal radiation by the clouds during the stratus phase creates a cooling at the cloud top as demonstrated
by the profiles evolution of the Long-Wave Heating Rate (LWHR) at Figure 7h. The more numerous are the cloud droplets the stronger the cooling is, as shown in the figure that SWHR can reach $-50\,\mathrm{K\,day^{-1}}$. This strong longwave emission is able to reduce the thermal production of turbulence at the cloud top and contribute to the stability of air masses at the cloud top, deepening the temperature inversion above cloud top. A stabilized cloud top layer by radiative cooling and a NLLJ core contributing to the shear-driven turbulence below the cloud base allows the well mixing of the cloud layer, making the LCL
to correspond to the LLCSs base as seen at Figure 2 (Adler et al., 2019; Lohou et al., 2020). In fact, modeled cloud layers are very stable before the sunrise as observed at Savè during the DACCIWA campaign.





### 3.2.2 Stratus phase

The stratus phase starts just after the sunrise. To maintain stratus in almost the same state as in the previous period needs certain proper temperature and humidity conditions as shows at Figure 8. Indeed, the temperature at the ground is still at 23°C at

06:00 and 08:00 UTC, and 20°C at the mean CTH (500 and 550 m respectively). RH profiles indicate that supersaturation still exists between the range of CBH and CTH, allowing droplets condensation to continue. Air masses are quite stable between ground and CTH during this time as the equivalent potential temperature ($\theta_e$) is near equal to 347 K and the inversion layer is settled where $\theta_e$ is reaching $350 - 351$ K. The horizontal wind speed between the ground and the cloud base decreases, which indicates the reducing of the NLLJ core (nearly $2 \mathrm{~m~s}^{-1}$) and its rises in altitude due to the turbulent mixing induced by the

LW cooling at the cloud top during the night. The turbulence between ground and cloud middle decreases to $0.03 \mathrm{~m}^2 \mathrm{~s}^{-2}$ then finally increase slightly to $0.04 \mathrm{~m}^2 \mathrm{~s}^{-2}$ at the mean CTH. The TKE is a bit stronger at 08:00 UTC, reaching $0.05 \mathrm{~m}^2 \mathrm{~s}^{-2}$ in the cloud layer which is explained by an increase of the vertical wind speed.

The aerosol concentration at 06:00 and 08:00 UTC is around $2000 \mathrm{~cm}^{-3}$ up to 500 m then it decreases along altitude. This

concentration is still high to allow the formation of $1100 - 1200 \mathrm{~droplets~cm}^{-3}$ between CBH and CTH. The concentration of cloud droplets leads to a maximum droplet radius of 6 μm, which is still not enough to form drizzle. The cloud layer has an albedo close to 1 due to the high droplet concentration. The presence of light absorbing aerosol causes the Short-Wave Heating Rate (SWHR) amplification at the cloud top by semi-direct effect. At 08:00 UTC, the SWHR and LWHR are equal to $27 \mathrm{~K~day}^{-1}$ and $-70 \mathrm{~K~day}^{-1}$, respectively.

At 10:00 UTC, the cloud layer starts to rise significantly, with CBH and CTH reaching 340 and 660 m respectively. Moreover, more solar radiations can reach the ground ($220 \mathrm{~W~m}^2$), leading to the heating of the surface and the increasing of the sensible and latent heat fluxes as seen in Fig 6. It also increases the temperature near ground to $24 \degree$C and at the cloud top the temperature is $20 \degree$C. Supersaturation occurs obviously between cloud base and cloud top and the inversion layer is observed

above the cloud top between 660 and 750 m. The NLLJ core is no longer present as the horizontal wind speed decreased to $0.5 - 0.75 \mathrm{~m~s}^{-1}$. However, the TKE increases to $0.1 \mathrm{~m}^2 \mathrm{~s}^{-2}$ almost to the cloud top. This enhancement of turbulence is expected to increase entrainment from above entering the cloud. Cloud droplets number concentration reaches a maximum of $1000 \mathrm{~droplets~cm}^{-3}$ and a maximum droplet radius reaching near 6 μm. Given these values, the cloud albedo is near equal to 1 and the SWHR increases to $45 \mathrm{~K~day}^{-1}$. It almost compensates the LWHR value of $65 \mathrm{~K~day}^{-1}$. During the 3rd of July, the

radar detected light precipitation from higher clouds but this simulation does not model this type of clouds. So, the cloud deck stands longer due to the set-up of tendency profiles especially the one for humidity which keeps supersaturation at stratus layer until 10:00 UTC.





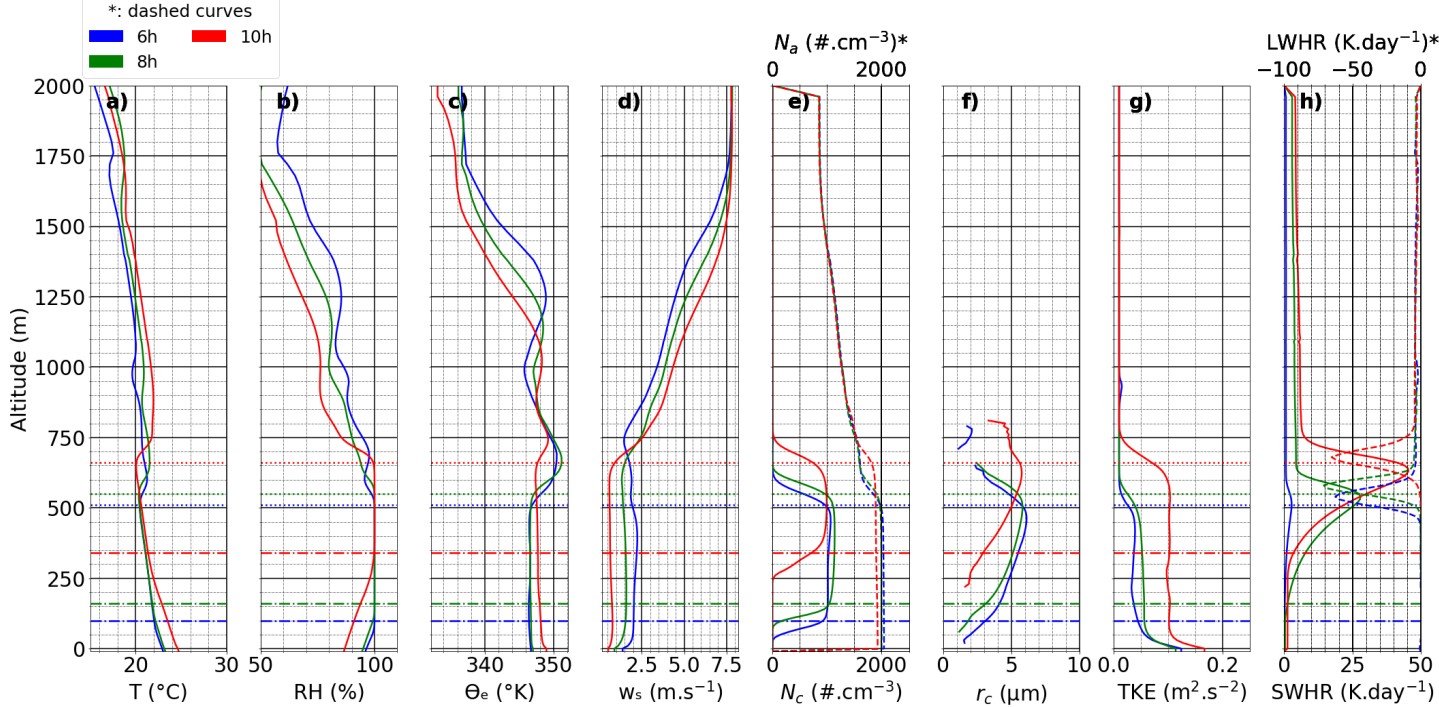

**Figure 8.** Profiles from left to right of temperature (T, a), relative humidity (RH, b), equivalent potential temperature ($\theta_e$, c), horizontal wind speed (w_s, d), aerosol number concentration ($N_a$, dashed curve, e), cloud droplets number concentration ($N_c$, plain curve, e), cloud droplet radius ($r_c$, f), turbulent kinetic energy (TKE, g), longwave heating rate (LWHR, dashed curve, h) and shortwave heating rate (SWHR, plain curve, h) at 06:00, 08:00 and 10:00 UTC. Dashdot horizontal lines represent mean cloud base height (CBH) and dotted horizontal lines the mean cloud top height (CTH).

### 3.2.3 Convective phase

This phase is extending from 12:00 to 17:00 UTC on 3 July 2016 case when the SW radiation flux at surface is maximum at 300 W m$^2$ (Figure 6), leading to a more intense heating from the surface. During this period, the temperature at ground evolves from 25 to 27 °C as seen at Figure 9. The temperature near the cloud top is 20 °C at 12:00 and 14:00 UTC. At the break-up time (16:00 UTC), the temperature is lower than 18 °C at the CTH. The formation of clouds is possible due to RH exceeding 100% in upper altitude, as the convection of humid air masses causes the CBH and CTH to rise from 450 to 925 m and from 760

to 1100 m, respectively. Moreover, at the break-up (16:00 UTC), $\theta_e$ decreases above 450 m of altitude, indicating air masses become more unstable with altitude. The horizontal wind speed is weak at the beginning of the phase with 0.5 m s$^{-1}$ at ground level but increases along time to reach 1 m s$^{-1}$ at ground and 3 m s$^{-1}$ around 700 m. This increase indicates the end of clouds break-up and the arrival of the marine inflow.





The turbulence profiles evolve along altitude during the convection, reaching 0.075 m² s⁻² in center of clouds and almost

zero near 850 m at 12:00 UTC. This profile evolves at 14:00 UTC to reach 1.25 m² s⁻² and zero at cloud center and 1000

m, indicating a reinforcement of turbulence due to an elevation of dynamical production via the vertical wind speed increase.

Finally at break-up time, turbulence near the ground decrease but at cloud level the TKE has a value of 1.5 m² s⁻² showing a

strong turbulence layer. This turbulent layer further moves to a upper altitude after 17:00 UTC.

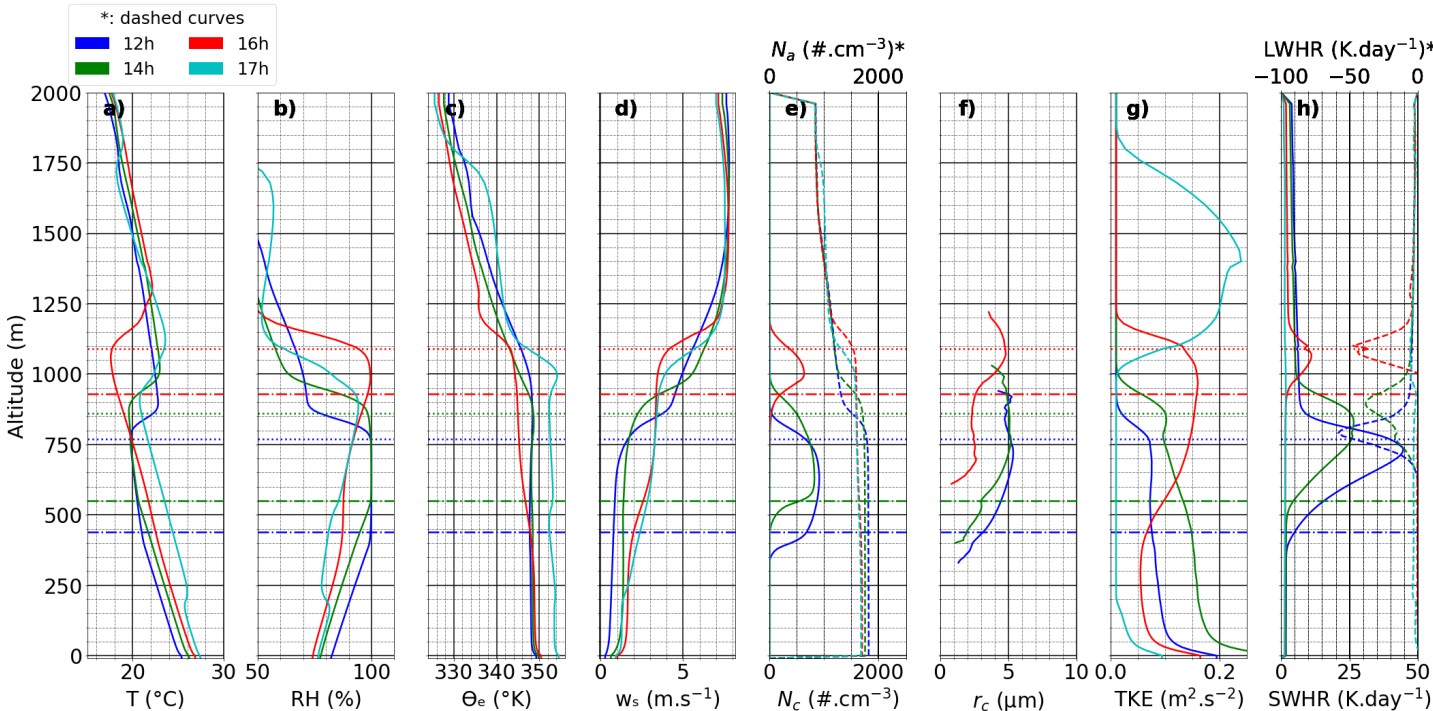

**Figure 9.** Profiles from left to right of temperature (T, a), relative humidity (RH, b), equivalent potential temperature ($\theta_e$, c), horizontal wind

speed (w_s, d), aerosol number concentration ($N_a$, dashed curve, e), cloud droplets number concentration ($N_c$, plain curve, e), cloud droplet

radius ($r_c$, f), turbulent kinetic energy (TKE, g), longwave heating rate (LWHR, dashed curve, h) and shortwave heating rate (SWHR, plain

curve, h) at 12:00, 14:00, 16:00 and 17:00 UTC. Dashdot horizontal lines represent mean cloud base height (CBH) and dotted horizontal

lines the mean cloud top height (CTH).

The aerosol distribution varies along with the dynamical situation. The maximum aerosols concentration reaches 1800 cm⁻³

below 800 m and 1700 cm⁻³ below 1000 m at 12:00 and 17:00 UTC respectively. Again with so numerous aerosols, the cloud

droplets concentration as a domain mean has a maximum value of 900 droplets cm⁻³ at 12:00 UTC. This value decreases

along time as more clouds dissipated. At the break-up of clouds, reduced cloud coverage allows more solar radiation to reach

the ground. The maximum value of SWHR drastically changes from 45 K day⁻¹ at 12:00 UTC (almost compensating cloud

top cooling) to about 10 K day⁻¹ at 16:00 UTC. The cloud top cooling is near constant at the end of convection phase with

−45 K day⁻¹.





## 4 Sensitivity study to examine the influence of different aerosol profiles on LLSC diurnal cycle

### 4.1 Aerosol profiles used in sensitivity simulations

The result of REF simulation has demonstrated that the Meso-NH model is able to reproduce many observed features of the
435 3rd of July LLSC case even though Meso-NH model has some biases. Moreover, the dynamical, thermodynamic and aerosol parameters are reasonably well simulated by the model. It is well known that aerosol, from both anthropogenic activities and biomass burning emissions, may influence cloud formation directly through absorbing solar radiation and indirectly by serving as CCN. To identify the aerosol sources and key processes of aerosol in such cloud enhancement, we tested the respective influence of anthropogenic sources and aerosol semi-direct effects. For this purpose, we have configured two different aerosol
scenarios based on observations during the field campaign (Figure A1 and Table 2), then applied them in a set of sensitivity simulations that would be otherwise the same as the configuration of REF simulation.

| Case | | $N_a$ (cm$^{-3}$) | $\sigma$ | D (nm) |
|---|---|---|---|---|
| POL | Mode 1 | 17100 | 1.54 | 55.19 |
| | Mode 2 | 2650 | 2014 | 101.83 |
| CLEAN | Mode 1 | 65 | 1.49 | 63.98 |
| | Mode 2 | 153 | 1.53 | 190.97 |

**Table 2.** POL and CLEAN aerosols size distribution described by two modes configured following three parameters (number concentration, standard deviation and diameter).

To investigate the impacts of anthropogenic and biomass burning sources on clouds, three additional numerical experiments were performed in addition to REF (Table 2): (1) an experiment with strong anthropogenic pollution influence based on the
445 aerosol chemical composition observed by Brito et al. (2018) and Denjean et al. (2020a) in urban plumes originating from the polluted cities of Lomé, Accra and Abidjan, which is named POL; (2) an experiment designed to underestimate aerosol emission by deriving REF aerosol concentration by 10, named CLEAN; and (3) an experiment without (with) aerosol semi-direct effect, called ADEOFF (ADEON) .

### 4.2 Impact of aerosol loads on microphysical and macrophysical properties of low level clouds

Figure 10 compares three experiments conducted with enhanced anthropogenic emissions (POL), underestimated aerosol emissions (CLEAN) and background conditions (REF) to provide a quantitative estimation of aerosol loads on radiation and LLSCs. In these simulations, both the semi-direct and the indirect effects are taken into account, which act simultaneously on cloud formation and evaporation. The POL case is mostly similar to REF. Before the sunrise, the mean CBH and CTH and cloud presence probability of both cases are almost the same until 08:00 UTC (see Figure 4 and Figure A2a and A2b). After
this time, the POL mean CBH is 10 m inferior to the reference reaching 340 m at 10:00 UTC while the mean CTH is mostly 10 superior even reaching 940 m instead of 920 m at 15:00 UTC. The cloud presence probabilities of both cases are also largely





the same until the break-up stage. For example, the cloud extent reaches 670 m (from 630 to 1300 m above the ground) at 16:00 UTC instead of 540 m in REF. On the other hand, POL and REF have produced clearly different cloud droplet number concentrations alongside mean radius throughout the life time of modeled clouds (Figure 10a and 10b). At the cloud formation

(02:00 UTC), despite having similar LWC around $0.35\,\mathrm{g\,m^{-3}}$ at 250 m in both cases, $N_c^{POL}$ reaches 333 droplets $\mathrm{cm^{-3}}$ with $r_c^{POL}$ of 6.45 μm instead of 653 droplets $\mathrm{cm^{-3}}$ and 5.1 μm for REF case. This is explained by the differences in the characterization of the aerosols between the two scenarios and the vertical wind speed as Abdul-Razzak and Ghan (2000) include vertical wind speed in their activation scheme. At 02:00 UTC this parameter is less than $0.30\,\mathrm{m\,s^{-1}}$ and allow POL and REF clouds droplets concentration and radius to evolve this way. This trend is reversed the droplets number concentration and radius

at 06:00 UTC these numbers are equal to 1208 droplets $\mathrm{cm^{-3}}$ and 6.43 μm for POL and 1305 droplets $\mathrm{cm^{-3}}$ with a radius of 6.12 μm for REF, respectively. After 08:00 UTC and until the clouds break-up, $N_c^{POL}$ is superior to $N_c^{REF}$ by reaching a maximum difference of 1425 droplets $\mathrm{cm^{-3}}$ at 14:00 UTC. Their respective radius are 4.42 μm and 5.18 μm while the LWC profiles are quite the same near $0.47\,\mathrm{g\,m^{-3}}$ at 750 m. These results are in good agreement with the ACPIM parcel model simulation done by Taylor et al. (2019) where $N_c$ varies in a range of $500 - 1400$ droplets $\mathrm{cm^{-3}}$ depending on the inland or

offshore (offshore + local emissions) aerosols origin.

The difference in cloud macrophysical features such as CBH and CTH between CLEAN and REF are visible though largely limited to a few tens of meters. However, their differences in other macrophysical features including cloud coverage and microphysical features are rather significant. Indeed, from formation to break-up of stratiform clouds, $N_c^{CLEAN}$ is inferior to $N_c^{REF}$

and $r_c^{CLEAN}$ is superior to $r_c^{REF}$. At 02:00 UTC, $N_c^{CLEAN}$ has a maximum value of 181 droplets $\mathrm{cm^{-3}}$ for a radius of 7.58 μm instead of 653 droplets $\mathrm{cm^{-3}}$ and 5.1 μm for $N_c^{REF}$ and $r_c^{REF}$ respectively with the same LWC value ($0.35\,\mathrm{g\,m^{-3}}$). Between 02:00 UTC and 08:00 UTC, $r_c^{CLEAN}$ increases to reach at the latter time 12.55 μm. After 08:00 UTC, $r_c^{CLEAN}$ decreases slowly showing a maximum value of 10.97 μm at 14:00 UTC. It has to be notified for this time that $LWC^{CLEAN}$ reaches near $0.45\,\mathrm{g\,m^{-3}}$ instead of $0.49\,\mathrm{g\,m^{-3}}$ for $LWC^{REF}$. This change in the available volume of liquid water with such

droplet size explains the decreasing of $N_c^{CLEAN}$ while the aerosol number concentration remains stable around $200\text{–}250\,\mathrm{cm^{-3}}$.

As demonstrated from previous discussions, that modeled cloud microphysical features respond to the variation of aerosol number concentration as expected, i.e., higher aerosol concentration leads to higher cloud droplet number concentration (POL > REF > CLEAN) while smaller mean droplet radius (POL < REF < CLEAN) and hence a higher cloud reflectivity (POL >

REF > CLEAN). However, interestingly, as shown in the bottom panel of Fig. 10, the response of the incoming solar radiation (SWRADSURF) at ground does not follow always such an expectation. In fact, SWRADSURF appears to be higher in POL than REF from sunrise to 13:00 UTC, and the values in both runs also clearly higher than that in CLEAN. This tendency is only reversed after 13:00 UTC when solar flux reaches its peak until break-up stage.







**Figure 10.** Evolution of cloud droplets concentration $N_c$ (top) and cloud droplets radius $r_c$ (middle) with the scenarios given and designated by letter a (REF), b (POL) and c (CLEAN). Bottom panel gives the evolution of mean domain SWRADSURF differences between POL / CLEAN and REF.

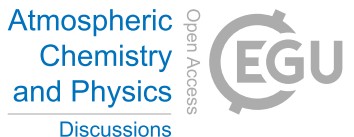

**Figure 11.** Liquid water path (LWC, $g\,kg^{-1}$ m) in POL (left column), REF (mid-column), and CLEAN (right column) ADEON runs at 10:00 UTC (top row), 14:00 UTC (middle row), and 16:00 UTC (bottom row)





Figure 11 shows that the major reason behind the above-described trend of SWRADSURF is the difference in cloud coverage in competing with the effect brought by different cloud reflectivity of various runs, especially before the noontime when zenith angle is still low. After sunrise, the cloud top starts to rise and cloud layer becomes thicker. In the meantime, this upward development brings a downward entrainment of dry air from the inverse layer above cloud top and causes evaporation in the cloud. For cloud with large quantity of very small droplets as in POL and REF, the evaporation would exceed more rapidly than in CLEAN case, thus cloud-void or thin cloud layer would form much easier than in the latter case. As shown in Fig. 11 and Table 3, cloud layer in CLEAN is slightly denser than those in POL and REF while cloud-void or thin cloud pixels account a substantially lower ratio within the domain. When direct solar flux is relatively low, cloud reflectivity seems becoming the secondary factor comparing to the cloud-void space in determining the value of SWRADSURF. As a result, SWRADSURF in CLEAN is significantly lower than REF then POL until zenith angle becomes higher closer to noontime. This would also have reduced the turbulent mixing as well as delayed the convection. At 14:00 UTC, difference in cloud thickness and cloud-void space still exists while becomes relatively smaller among the three different runs (Fig. 11 and Table 3), cloud reflectivity now becomes the primary reason to cause a different SWRADSURF as shown in Fig. 10 (bottom panel). Interestingly, modeled clouds in POL and REF appear to dissipate earlier and much faster than in CLEAN in the break-up stage (Fig. 11, bottom panel), the very low number of cloud pixels with different thickness in the domain seems having brought some variations in SWRADSURF.

|       | LWP 10 UTC | PCP 10 UTC | LWP 14 UTC | PCP 14 UTC | LWP 16 UTC | PRP 16 UTC |
|-------|------------|------------|------------|------------|------------|------------|
| POL   | 14.87      | 12.79      | 10.98      | 42.17      | 1.96       | 99.66      |
| REF   | 15.67      | 10.11      | 11.34      | 42.69      | 2.74       | 99.67      |
| CLEAN | 16.98      | 6.95       | 11.79      | 44.93      | 4.12       | 94.47      |

**Table 3.** Domain averaged liquid water path (LWP, $g\,kg^{-1}\,m$) and poor-cloud pixel percentage (PCP, defined by the percentage of pixels where LWP < $10\,g\,kg^{-1}\,m$; percentage) in three different runs.

To summarize, as expected, aerosol concentration is a major factor in controlling the cloud microphysical features by changing the simulated droplet number concentration and radius for clouds with similar liquid water content. However, despite this well-known Twomey effect the incoming solar radiation at ground did not decrease due to additional $N_c$. Instead, cloud macrophysical features and in particular cloud-void space caused by dry entrainment from inverse layer above the cloud is the dominant factor impacting the incoming solar radiation at ground. Cloud macrophysical properties determine the break-up speed of modeled clouds and therefore, the life cycle of the modeled LLSCs. It is worth indicating though, another factor that might contribute to the cloud life cycle, i.e., the atmospheric heating caused by the semi-direct optical effect of absorbing aerosol component such as black carbon has not been analyzed up to this moment and will be discussed in the following section.





### 4.3 Impact of aerosol semi-direct effect on low level clouds

The semi-direct effect of aerosols on LLSCs that represents the modifications of the LLSCs properties and atmospheric
dynamics due to absorption of SW radiation by absorbing aerosol, has been estimated by conducting three additional experi-
ments constructed accordingly as same as the original experiments (hereafter ADEON including REF, POL, and CLEAN) but
excluding aerosol direct effects (named ADEOFF) and then comparing the results between each of the paired runs. Indeed,
BC is the major species behind the semi-direct effect in our case study as proved by scenario REF_NOBC as seen in Figure
A2c. This scenarios retakes the characteristics of REF in term of size distribution and total mass but replace BC mass part
by a equivalent mix of sulfate, ammonium, nitrate and organic carbon. By running ADEON and ADEOFF configurations for
REF_NOBC and comparing cloud evolution and cloud presence probability, it can be seen that there is few minor differences
between these two configurations.

The changes in cloud vertical dispersion, SWHR and TKE due to aerosol absorption and potential feedbacks are shown in
Figure 12. The results demonstrates that light-absorbing BC aerosols can cause a substantial atmospheric heating accompanied
by substantial warming tendency near LLSCs-top (Figure 12b). The domain averaged heating due to BC aerosols (SWHR) in
CLEAN case is rather insignificant in comparison with the two other cases as $1.30 \, \mathrm{K \, day^{-1}}$ (and a cooling $-2.25 \, \mathrm{K \, day^{-1}}$
above due to cloud top change) at 14:00 UTC, whereas it reaches $12.16 \, \mathrm{K \, day^{-1}}$ $(-13.14 \, \mathrm{K \, day^{-1}})$ for POL, $7.71 \, \mathrm{K \, day^{-1}}$
$(-9.24 \, \mathrm{K \, day^{-1}})$ in REF, respectively. Accordingly, in ADEON runs, more water vapor tends to condense onto cloud droplets
under the higher relative humidity in the lower PBL and decreasing turbulent mixing (Figure 12c, with a maximum decreasing
of $-0.18 \, \mathrm{m^2 \, s^{-2}}$ for POL) lead to a decrease of the cloud top height and also reduced reflection due to BC in-cloud absorption.
The cloud top height reduction in two polluted cases POL and REF is quite significant as shown in Figure 12a, where CTH
in POL and REF has decreased by up to 100 and 70 meters due to the presence of BC, respectively. On the other hand, CBH
is also increased about 20 meters in both cases before break-up. In comparison, CTH and CBH appear to be less affected in
CLEAN run due to its low BC content. Before break-up, in-cloud TKE below the heating layer has been reduced in some extent
(Fig.12c). On the other hand, due to a lower cloud top in polluted cases, boundary layer top with active turbulent exchange
would also be lowered. The effect of BC absorption in lowering modeled clouds in POL and REF (implying a reduced upward
development) is likely another factor to slow down their break-up as discussed before.

We find that the semi-direct effect can manifest both an enhancing and a weakening contribution to the (negative) indirect
radiative forcing (Lohmann and Feichter, 2001; Koch and Del Genio, 2010a; Huang et al., 2014; Yamaguchi et al., 2015;
Stjern et al., 2017; Kreidenweis et al., 2019). At 14:00 UTC, the flux difference between ADEON and ADEOFF at ground
reaches $-33 \, \mathrm{W \, m^{-2}}$ and $-75 \, \mathrm{W \, m^{-2}}$ for REF and POL respectively (Fig. 13c). This is explained by a decreased void space
in ADEON runs that allows less solar radiations to attain the surface despite the cloud layer being thinner (Fig. 12c and A3),
hence the semi-direct effect contributes positively to the enhancement of (negative) indirect radiative forcing in this case.







**Figure 12.** Evolution of the difference of the mean CBH and CTH (a), SWHR (b) and TKE (c) between the simulation runs with and without aerosol direct effect (ADEON-ADEOFF) for REF, POL and CLEAN.

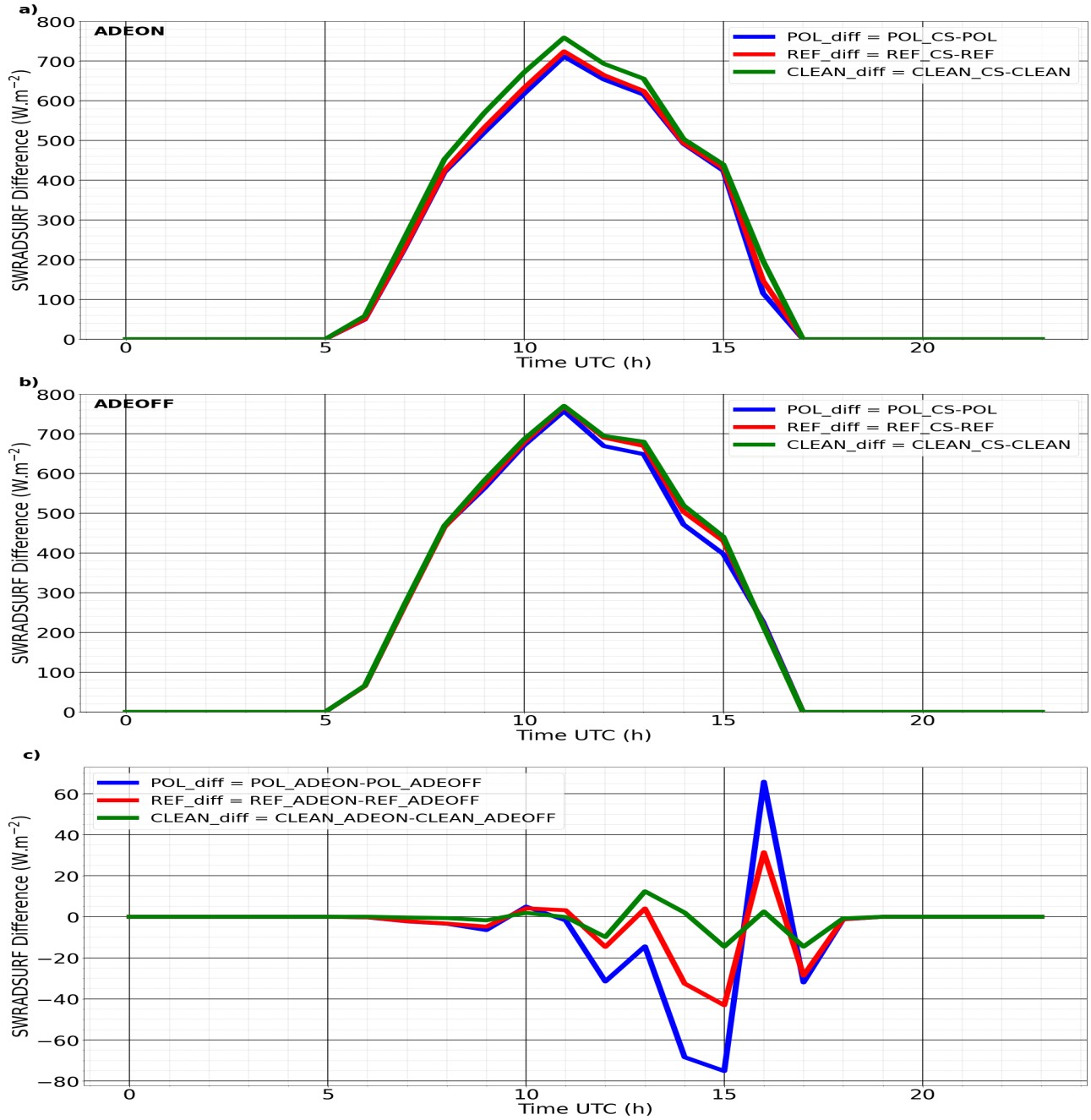

**Figure 13.** Mean difference surface SW radiative flux (SWRADSURF) between Clear-Sky (CS) and cloudy scenarios giving the flux dissipated by clouds in ADEON (a) and ADEOFF (b) configurations. SWRADSURF difference between ADEON and ADEOFF configuration for the three scenarios (c).


But at 16:00 UTC the flux difference between ADEON and ADEOFF becomes positive with values for REF and POL as 32 W m$^{-2}$ and 66 W m$^{-2}$, respectively. As the clouds break-up slower in ADEOFF during this stage due to thicker clouds,
more clouds inside the domain with increased thickness causes less SW radiations reaching ground. In this case, the semi-direct effect weakens the indirect radiative forcing.

The above results have demonstrated the important role of solar absorption by aerosols in determining the life cycle of LLSCs. The atmospheric heating by light absorbing BC would limit the elevation of cloud layer, especially during the break-
560 up stage (Koch and Del Genio, 2010b; Zhang and Zuidema, 2019). Such heating can also decrease cloud-void space then delay break-up until late afternoon, especially for clouds with higher cloud droplet number concentration in polluted environment such as in POL and REF runs. This study case also exhibits either a positive (e.g., decreasing cloud-void space) or a negative (e.g., accelerating break-up in late afternoon due to a thinner cloud) contribution of the semidirect effect to the indirect radiative forcing.

**5   Conclusions**

A characteristic case of LLSCs over SWA have been simulated with Meso-NH model in high-resolution Large-Eddy Simulation configuration constrained by DACCIWA measurements. The model has successfully reproduced observed life cycle alongside key macro and microphysical features as well as surface radiative and heat fluxes. To determine the impact of aerosols on the LLSCs diurnal cycle, sensitivity simulations using several different aerosol profiles have also been conducted.
These aerosol profiles contain different number concentrations and chemical compositions in order to reflect the situations associated with various aerosol populations encountered during the campaign.

The results from various sensitivity simulations suggest that both aerosol concentration and chemical composition can ef-
575 fectively influence the LLSCs life cycle. The impact of the aerosol concentration, as reflected from a comparison among simulations using aerosol profiles with different number concentrations, is initiated from resultant cloud microphysical features in particular the cloud droplet number concentration and mean droplet size. Such a difference created by different aerosol number concentration also affect cloud reflectivity as expected. Interestingly, we have found that the difference in cloud reflectivity caused by different aerosol concentration does not always dominate the surface incoming solar radiation and thus
cloud development after sunrise due to another competing factor: the cloud-void space caused by the air entrainment from the inversion layer above cloud top dominates the variation of surface incoming solar radiation before noontime. Clouds influenced by higher aerosol concentrations and thus higher cloud droplet number concentration while smaller droplet sizes are found to evaporate more easily and thus impose more cloud-void spaces. For the same reason, clouds with higher droplet concentration are likely to break-up earlier.

In addition, our sensitivity runs including versus excluding aerosol direct radiative effects have also demonstrated the impact specifically of solar absorption by black carbon on the cloud life cycle. The excessive atmospheric heating up reaches 12 K day$^{-1}$ in our modeled cases by black carbon can lower the cloud top height and reduce dry entrainment. Working with above-indicated aerosol concentration effect this might delay break-up until late afternoon, while beyond that modeled clouds in polluted cases with higher aerosol concentrations and BC included would break-up faster due to thinner cloud layer. Therefore, semi-direct effect can contribute positively to the indirect radiative forcing (negative in quantity) due to decreased cloud-void space, or negatively by causing thinner cloud layer and thus a faster cloud break-up in late afternoon, all depending on the phase in stratiform cloud diurnal cycle.

Our study has demonstrated that the life cycle and thus the radiative forcing of LLSCs over land area can be substantially influenced by aerosol including long-range transported biomass burning and locally emitted urban plumes. In fact, more aerosol profiles had been collected during the DACCIWA campaign besides the few used in this study. Future research works could reveal the aerosol impact under an even broader range of aerosol properties and to examine the temporal variations of LLSCs radiative effects evolved with different large-scale meteorological conditions with different associated airmass. More analysis on different clouds situation in SWA would also be able to assess or refute current results on semi-direct effect.

*Code and data availability.* The data obtained during the DACCIWA campaign at the Savè supersite are available on the SEDOO database (http://baobab.sedoo.fr/DACCIWA/). The Meso-NH code is maintained and updated by LAERO and CNRM, it is freely available for download at http://mesonh.aero.obs-mip.fr/mesonh52/. All data used in this study are publicly available on the AERIS Data and Service Center, which can be found at http://baobab.sedoo.fr/DACCIWA.

*Author contributions.* LD and CW designed the simulations and LD conducted model simulations and data analyses. LD wrote this paper with contribution from all co-authors. CW advised and helped to better understand the different aspects of this research work. PT advised and trained first author for Meso-NH and ORILAM module use. CD processed and provided the aerosol profiles used in previous simulations and NM was part of this work. MZ helped to select the study case and advised during the study case construction and analysis. AD brought a critical eye to this work.

*Competing interests.* The authors declare that they have no conflict of interest.





*Acknowledgements.* This study is supported by L'Agence National de la Recherche (ANR) of France under "Programme d'Investissements
d'Avenir" (ANR-18-MPGA-003 EUROACE) and co-funded by University Toulouse III Paul Sabatier. The computation of this work was
performed using HPC resources of French GENCI-IDRIS (Grant A0110110967 and A0090110967) and French Regional Computations
center CALMIP. The first author thanks the Laboratoire d'Aérologie, Université de Toulouse, France, for hosting the research activities. A
special thank to all people whose work was involved in the measurement and processing of DACCIWA campaign data especially over the
Savè supersite. First Author thanks Quentin Rodier from CNRM, France, for his advice on Meso-NH use. Benoit Vié and Marie Mazoyer
(CNRM) are thanked for their help to handle and modifiy microphysical scheme LIMA. A thank to Quentin Libois (CNRM) for bringing
its understanding of Meso-NH's radiative schemes. Many thanks for all co-authors for their help and time given to this article and a special
thanks to Fabienne Lohou (LAERO).

## Appendix A

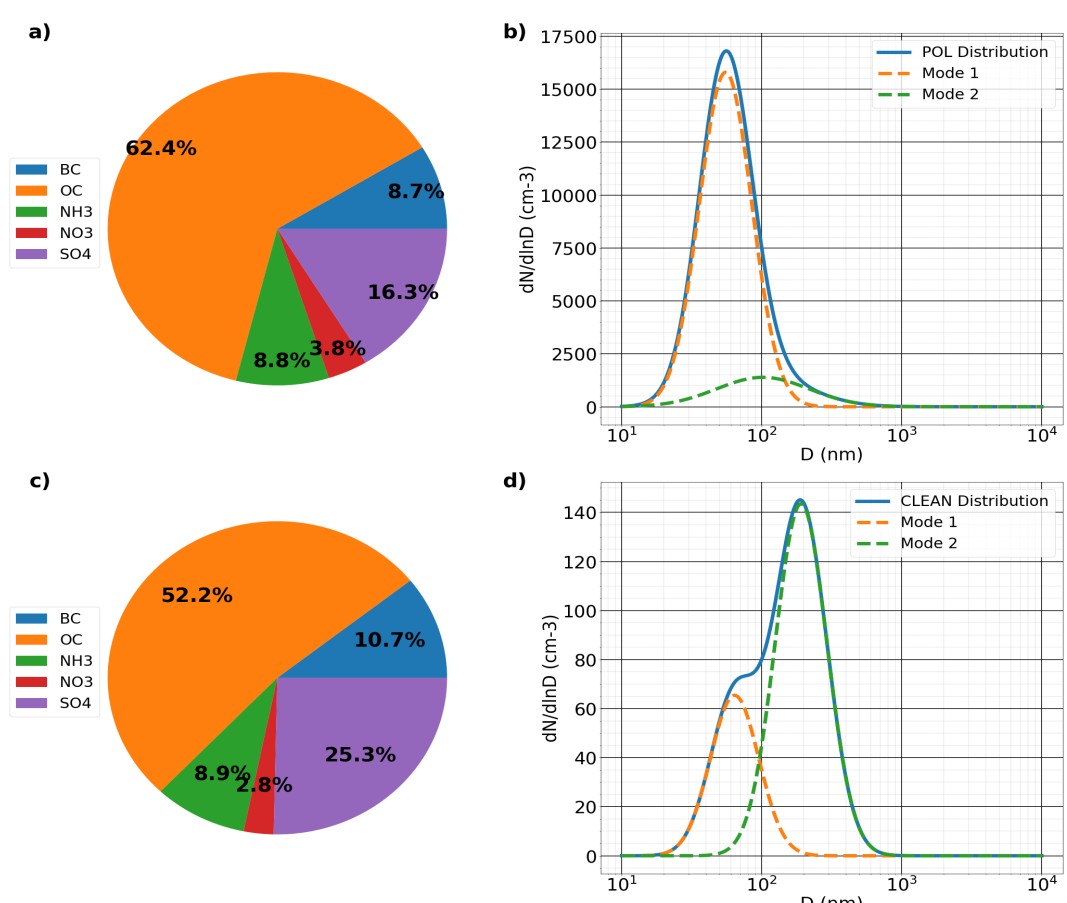

**Figure A1.** Mass composition (a,c) and size distribution provided by (Denjean et al., 2020a) and fitted into 2 modes described in Table 2
(b,d) for scenarios POL (top), CLEAN (bottom).





**Figure A2.** Mean LLSCs deck evolution of POL (a) and CLEAN (b) cases with the representation of REF's one to make comparison and REF_NOBC ADEON and ADEOFF runs (c), vertical colored bars for POL/CLEAN (left) and REF (right) attribute at each altitude level a cloud presence density for both cases at each hour.







**Figure A3.** Liquid water path (LWC, $\mathrm{g\,kg^{-1}\,m}$) in POL (left column), REF (mid-column), and CLEAN (right column) ADEOFF runs at 10:00 UTC (top row), 14:00 UTC (middle row), and 16:00 UTC (bottom row)



# References

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
