# Peer review of "The Impact of Aerosols on the Stratiform Clouds over Southern West Africa: A Large-Eddy Simulation Study"

_Atmospheric Chemistry and Physics, 2022_

## Referee Comment (RC1)

**Review of "The Impact of Aerosols on the Stratiform Clouds over southern West Africa: A Large-Eddy Simulation Study"**

**Summary**

This is a modeling work that uses LES for a coastal stratocumulus case study from the DACCIWA campaign in southern West Africa. The main focus of the work is given to the different effects of aerosols, for which three different aerosol size distributions are considered, which are then transformed to a total of six cases by switching off the black carbon content. Results highlight how spatial organization aspects of the cloud break-up play a role in the aerosol effects. Their effort in modeling a real case is quite valuable, and it gives them good results when compared to in-site observations. Their sensitivity analyses point at the different effects of aerosols, some of them unexpected, adding novelty to the work. Over all, the work is very interesting, the methods are good, and the topic is highly relevant. However, there are some aspects of the manuscript that should be improved prior to publication.

**Major comments**

- There are multiple writing mistakes along the document, along with confusing statements. Please revise the manuscript carefully. Some suggestions are given below, but still, readability is essential for a paper.

- Cloud break-up is used to refer to a state where the cloud deck has a low cloud fraction, but there is no clear definition of it nor a discussion about it. Some sentences talk about earlier or faster break-ups but I'd suggest to treat the concept carefully to avoid confusions, specially since coastal clouds go through cloud dissipation and the process is a complex one already.

- The case studied was said to be a late dissipation one. They hypothesized that precipitation evaporation could explain this. No further analysis was done using LES data related to this issue, I think it should be diagnosed. I wonder if studying a more usual dissipation time case would give the same results. A more critical comment could be appreciated.

**Minor comments**

- I'd suggest to simplify the Meso-NH model description, focus on the setup and how the data was used to create initial and boundary conditions, and leave the model details on Appendix if desired. In my opinion, later experiments on aerosol contributions can also be described in the model section. This is personal preference, as I've seen both around.

- Also related to readability, the results section can be summarized greatly, focusing on the big takeouts instead of carefully describing every value presented in the figures. While a description of the time evolution of the different vertical profiles seems fitting for this type of work, I'd suggest to be concise, highlight the most significant processes or values, since everything else is available in the plots.

- Cloud formation is said to occur around hour 2, which is probably already part of the LES spin-up time. Could this be an issue?

- As results are presented, there is no critical comparison with the references given in the introduction for similar processes. It is important when summarizing to compare and also state what is novel.

- Time-series of cloud fraction and liquid water path, both said to be available from observations and the LES runs, could complement the analysis greatly. Please include them in one of your figures, and strengthen the physical description of the PBL processes as needed.

- The cloud presence probability, CPP, is not carefully defined, so I don't understand how to interpret the values.

**Line by line comments**

- L10 Is it necessary to remark that there is no precipitation like this in the abstract? Maybe combine with previous sentence "...effect, with all cases having negligible precipitation".

- L15 Why use cloud-void instead of cloud fraction or cloud cover? This is true for all the document

- L21 What do you mean by stabilize? A more steady evolution?

- L23 Break-up faster means an earlier breakup time?

- L64 Why would aerosol emissions impact the wind?

- L74 Are these 2 studies observational or model-based?

- L80 It might be good to summarize what the direct, semi-direct, and indirect effects are, maybe in the second paragraph of the Introduction.

- L74 This paragraph has a lot of info but it's hard to follow as there is no clear story in it. The last sentence helps, but maybe it'd be good to have a similar introductory sentence and then highlight the knowledge gaps as info is presented. Just a suggestion. Another idea is to remove all unnecessary mentioned results if they won't be used for comparison or contrast.

- Fig. 1 Maybe a proper reference is better than the link.

- L98 What local scale and fine resolution are you talking about?

- L116 Local time is also UTC for this location, right?

- L124 The info in this paragraph is a bit heterogeneous; some descriptions are very detailed and others not as much. Is the importance given to the details of the sky camera meaning that this data is more important than the rest? Also, a matter of preference, but it could be easier to follow with a consistent sentence structure like: first the instrument, then the data it generates (as done first with the ceilometer).

- L135 Aren't these fluxes included in the last sentence?

- L162 Turbulent mixing maintains cooling? You mean through downdrafts or are you referring to the cooling advection?

- L168 What is the scenario C?

- L170 What does break-up mean here?

- L192 Add reference instead of link.

- L243 This resolution is not particularly high for stratocumulus LES

- L247 Are these values used to prescribed turbulence related to the local observations in any way? Is it okay for the profile to not change over time when modeling a coastal case?

- L249 if the model was tuned for the site observations, it'd be good to include that in an appendix. Otherwise, what kind of parameters is this surface model taking in for this case?

- L251 Are these 2 s chosen for numerically advancing the LES? If so it's more than adequate for diurnal variations, and probably needed not for that reason but for keeping a good CFL number. It's not clear as is written.

- L254 Do you mean using those observations as the initial condition? What is the nudging timescale for the tendency profiles?

- L258 How did you combine all that data to produce vertical profiles?

- Fig. 3 Improve the description: mass composition of what? Percentage text in a) is overlapping. Is the dot needed in the units of the b) y axis label (and all other labels)?

- Table 1 I'd suggest putting the symbol for each parameter after mentioning them in the description.

- Fig. 4 and other figures are vertically shrinked. "Vertical colored bars" are actually dots. Do simulated values represent the horizontally averaged CBH? Why are they reported hourly when the LES has a smaller step size? What is the meaning of this probability?

- L273 Do you say "formation" because the initial state for the LES is cloud-free? If so, initial conditions should be stated carefully.

- L274 Is the mean CBH estimated as the horizontal average of points where the first cloudy grid point is located?

- L280 You can be more specific than "some period"

- L284 Could the difference also be due to prescribing a weak subsidence profile or due to enhancing entrainment by not having a very fine vertical resolution?

- L289 This aids the previous paragraph in saying that for 10:00-14:00 UTC, observed CBH is below the 0 level probability. I'd move this and start a new paragraph with the spatial results.

- L289 What does a CPP of 50% mean? Is it the geometric midpoint?

- L294 By visible camera do you mean sky images in the visible range?

- L295 "distinct cloud rolls" instead of "clear cloud rolls", so that clear is not confused with clear sky. It can also be noted that this feature is not observed in the sky image

- L296 What is the CPP at the mid distance? At the mean in-cloud height? I still don't fully get the meaning of CPP values, so I don't understand what is useful of this description

- L301 Rather than demonstrating the break-up, it evidences the already broken field. This comment is related to what is the definition of break-up.

- L303 Still, the camera at Savè shows a big portion of the sky completely clear. The LES does not reach that type of organization.

- L317 Do you mean that the difference is due to the tuning being done for more persistent cloud decks?

- L319 Why would that difference be reduced in the convective phase?

- L354 along what altitude?

- Fig. 7 $\theta_e$ has not been defined, and its units are just K, not °K. Put w_s in equation mode. Why the discontinuous lines in f)? Why the choice of plotting T, RH, and $\theta_e$ instead of just $\theta_e$ and $q_t$?

- L376 Does stable mean constant cloud thickness here? Be careful not to be confused with thermal stability.

- L383 is this a fixed threshold to find the inversion height?

- L416 Does more unstable mean signs of decoupling?

- L417 What does "end of cloud break-up" mean? That the clouds cleared up or that it's fully covered again?

- Fig 9. Why report at 17 h and not 18 h to follow the 2 hours spacing? w_s to formula. Why does TKE, RH and $\theta_e$ increase above the PBL at 17 h?

- L435 What biases are you talking about?

- L440 Do these cases represent extreme situations in the set of observations?

- L456 What is cloud extent? Cloud thickness?

- L464 What does "this way" mean? I'm guessing not by the model activation but the text is confusing.

- L460 Here and in other places, comparing numbers in a more descriptive way can help the reader. For example, "$N_c^{POL}$ reaches half of the droplet concentration of the REF case (333 vs 653 droplets per cm$^3$), with a slightly higher radius (6.45 vs 5.1 $\mu$m)."

- Fig 10 Why are there 2 cloud layers at hour 6? It might be useful to also have a plot of the standard deviation of surface SW irradiance to accompany the last panel.

- L473 I'd suggest to check if the difference in LWC is related to changes in cloud cover. Also, since the changes in CTH and CBH are equivalent to just a few grid points, dive into that discussion as well, what was the expected outcome? Are there both positive and negative cloud thickness feedbacks that may be canceling each other?

- Fig 11 It is hard to distinguish the clear portions in the last panel. Maybe you can set the zero values as NaN for plotting them in white, and mention that in the caption. I don't know if it's related to the wind at the time but POL seems to have less elongated structures at 14 UTC. What is ADEON?

- L491 When saying "major" reason, what is the other reason why the trend would be different from what you expected?

- L492,L500 The solar zenith angle should be lower near noon, right? Do you mean solar elevation angle?

- L498 I don't understand this sentence well. When you say "direct solar flux is relatively low" you don't mean the time of the day, right? Then you talk about cloud reflectance, which I'm guessing you interpret through LWP or LWC, saying it is a secondary factor. Does this mean that for clouds without full cloud cover (which would be the primary factor), then changes in reflectivity are also promoting the unexpected result for the POL case?

- L500 By "this", do you mean the reduced PBL heating due to a higher reflected SW at cloud top?

- L504 Though the images do suggest that some layers may dissipate earlier, it'd be better to include time-series plots of LWP and cloud fraction, maybe as panels b and c in Fig. 4.

- L505 The last sentence is confusing. It seems to relate a low number of variable pixels with variations in surface SW irradiance.

- Table 3 Is this LWP or LWC (units of LWP are typically g/m$^2$)? Why don't you put the % symbol instead of writing percentage again? Last column title shouldn't be PRP, right?

- L511 Do you think that a different type of variation could cause the more expected result?

- L523 You should explain the REF_NOBC case here (I'm guessing it has no BC, but were all the other aerosols kept?)

- L534 You could also include this mentioned difference in liquid water content in Fig. 12, because it's not shown.

- Fig. 12 It might be useful to include cloud thickness in panel a too. Why are the first hours skipped? If break-up time is important, it could be marked in these plots.

- L541 I don't completely understand the last sentences. If the CTH is kept nearly constant, dissipation could still occur due to other factors, are all of them unchanged? This could be diagnosed using your difference approach on a sort of budget terms (see van der Dussen 10.1175/JAS-D-13-0114.1 and Ghonima 10.1175/JAS-D-15-0228.1 works on LWP and cloud thickness budget equations, I'm not sure if there's work relating them to aerosol effects).

- L550 This being said, maybe the analysis could benefit from comparing not only the domain averaged SW fluxes but by separating the domain in cloud-void and cloudy portions, in order to quantify how much the low cloud fraction effect weights.

- L535 This decrease in TKE is very interesting. I'd interpret it as limiting entrainment.

- L554 How do you know that clouds break up slower? If you mean a state with greater or lower cloud cover, I think that's different from a break-up speed. Still, a cloud cover vs time plot could hint towards that.

- L555 Note that clouds are also larger for the ADEOFF cases at 16 UTC, this is relevant for cloud organization and solar variability.

- L559 This is a bit confusing. Are these the effect of having BC or of not having it?

**Typos/Writing suggestions**

- L45 models

- L61 growth?

- L81 break-up or breakup? (both seen)

- L107 the radiative budget

- L116 hours, "This site" instead of "It"

- L118 "(up to 1500 m above ground level)" instead of "reaching...". You could skip "a.g.l" since it's the only time it's used in the paper.

- L124 This is just personal preference but why not just say "At Savé" instead of supersite of Savé everytime?

- L133 "Radiation fluxes" or irradiance instead of "radiations"

- L141 remove "analyzed"

- L146 Also measured

- L147 Wind speed and direction

- L147 The last sentence is very lacking given all the details above, and reads too casual. Suggestion: "A gas concentration analyzer was used to measure CO2, CH4, and CO content."

- L150-151 reads weird, maybe it's better a sentence of its own to talk about the NLLJ

- L157 Why ABL now? You had used PBL before. Pick one

- L153 were located

- L159 add comma after coast

- L165 identified three evolution scenarios

- L168 evolve

- L171 The co-located radar...

- L180 sulfates

- L231 grid point

- L234 layers... separated in up to...

- L238 of the observed

- L255 radiosondes instead of radiosounding profiles

- L258 resulting instead of resulted?

- Table 1 described instead of describes?

- L266 Maybe "nitrates" to be consistent with the plural

- L266 has been

- Fig. 4 "represent" instead of "represents"

- L273 formation of the observed LLSCs

- L282 the simulated mean CBH approaches again the ceilometer readings

- L291 "column, calculated ...pixels, ..."

- L291 Say that LWP is shown in Fig. 5.

- L292 nearly equal to

- L293,L295 observations

- L295 Is is 6:00 or 06:00 as in Fig. 2? Be consistent

- L298 "less homogeneous" instead of "more inhomogeneous"

- L298 "sky camera" instead of "cloud camera"

- L299 "clear" instead of "no-cloud", "the middle row"

- L300 "continues", "the convection phase"

- L303 the LWP map (Fig 5b)

- L302 "cloud thinning" instead of "decreasing cloud thickness", "the cloud deck"

- L305 "at the surface", ", averaged.. domain,"

- L309 follows the observations well

- L312 cloud deck

- L313 in this case, I support "break up" since it's used as a verb

- L314 observations

- L316 "very close at around", "break up"

- L317 during the DACCIWA

- L318 detected by the Savè

- L320 "of the modeled", "reproduced those measured by the instrument well". Remove "as shown in. . ." since you already mentioned which Fig.

- Fig. 6 I'd suggest replacing the dots as spaces in the label units as it was done in the caption.

- L333 fluxes

- L336, L362 microphysics

- L337 evolution

- L344 radiosonde

- L346 "increasing" instead of "that increase"

- L347 decreased

- L351 "greater than" instead of "more than", "cloud formation"

- L353 "which yields a well-mixed sub-cloud layer"

- L363 "droplets per cm$^3$", droplets in text mode

- L370 "The more numerous the cloud droplets are ..."

- L371 If saying "in the figure", say which one

- L379 "as shown in", "ground temperature"

- L381 " between ground and CTH": isn't that the whole PBL?

- L384 it rises

- L386 increases

- L392 You should define SWHR at the first mention in L371

- L397 "stronger solar irradiance" instead of "more solar radiations"

- L411 $W\ m^{-2}$

- L419 convection phase?

- L428 dissipate

- Table 2 Maybe you can merge some cells for a better table. The second value of sigma is very high, what are its units, or is it normalized? Maybe shortening CLEAN to CLN can help later for naming the other variables.

- L459 lifetime

- L464 Do you mean "reversed by"? Confusing sentence.

- L466 break up

- L467 radii

- L471 is visible

- L481 "those" instead of "that"

- L485 "Fig. 10c" instead of "the bottom panel of Fig.10"

- L487 runs are also

- L489 until the

- L492 before noon

- L495 For a cloud with a large quantity

- L495 Exceed what?

- L498 seems to become

- L521 "in the same way as". Also, you may want to split this sentence for clarity.

- L524 "These scenarios retake...", "in terms of", "replaces the BC mass..."

- L525 an equivalent

- L526 there are

- L530 demonstrate

- L531 by a substantial

- L532 Do you mean "(SWHR difference)"?

- L536 reduced SW reflection

- L549 solar irradiance

- L554 "break up" (used as verb), "due to being thicker"

- L555 weaker SW irradiance

- L559 Be more specific: cloud layer or cloud top?

- L564 semidirect or semi-direct?

- L566 has been

- L580 entrained

- L582 Is this sentence incomplete?

- L584 break up

- L590 "because" instead of "by"

- L591 What is that "this"?

- L592 break up

---

## Editor Comment (EC1)

The Impact of Aerosols on the Stratiform Clouds over southern West Africa: A Large-Eddy Simulation Study

In addition to the reviewers' comments, I have a number of important comments that I would like you to take into account before resubmission:

1) To begin with, the quality of the writing is not to up to par. I am sympathetic to the fact that the authors are probably not native English speakers but we cannot underestimate the importance of clearly written and sharp text that communicates the ideas effectively. Unfortunately, much work needs to be done in this regard.

2) On a scientific note, the authors do not seem to have strong familiarity with the literature on aerosol-cloud-radiation interactions and thus, while the results seem robust, some of the explanations are well-known and could be stated much more simply, with appropriate references. In many places this leaves the incorrect impression that the authors have discovered something new.

   Examples:
   - Liquid water path (LWP) and cloud fraction (CF) adjustments
     The notion that the Twomey effect is the dominant one has long been shown to be inadequate, especially given the much stronger control of N on LWP (2.5 x more important in a relative sense) and the dominance of CF, about which much less is known regarding aerosol effects. There is a very large body of literature on this topic and in various places, the text comes across as naïve (e.g., bottom of page 24, and bottom of page 25).
   - LWP adjustments are usually negative in stratiform clouds (https://doi.org/10.1175/1520-0469(2003)060<0262:TCALWT>2.0.CO;2 https://doi.org/10.5194/acp-19-5331-2019, doi:10.1029/2006GL027648
     This makes it a central variable for aerosol-cloud-radiation interactions, and yet other than the Table values, it's hard to get a good picture of LWP evolution and how N might be affecting LWP. The same is true for CF: the smaller the cloud fraction the smaller the radiative effect of the clouds, and the less leverage there is for aerosol effects on clouds. Aerosol effects on CF are less well quantified but this might be where some extra work gives you an opportunity to say something new.
   - On page 21, the discussion of the microphysical responses is long and not very informative because much is already anticipated. Is the POLLUTED case even needed given that REF is so polluted already and that at some point updraft/supersaturation production cannot activate any more aerosol?
   - Effect of drizzle: it has been proposed that weak drizzle can stabilize clouds (by preventing deepening https://doi.org/10.1175/1520-0469(1998)055<3616:LESOSP>2.0.CO;2) and if drizzle evaporates just below cloud base, can strengthen turbulence by destabilizing the BL (doi:10.1029/2001JD001502) When discussing drizzle, please engage in these ideas and see if they are relevant to your analysis (e.g., CLEAN, Fig. 10). In Fig. 10, a TKE profile would help to show

whether weak drizzle just below cloud base might be enhancing cloud turbulence/deepening. You could show divergence of the modeled drizzle flux to get a sense of evaporative cooling below cloud base.

- Where does the absorbing aerosol reside? This makes a significant difference to the dynamical response (e.g. doi: 10.1256/qj.03.61, doi:10.1029/2005JD006138, 10.1002/2015GL066544). And please convey the essence of knowledge already known from these papers, rather than simply providing lists of references. The current version of the text is not careful about using those references to provide context.

3) Missing information/other comments:
   - The cloud radar is mentioned but we aren't told its wavelength, which makes it hard to interpret what it sees. (See drizzle discussion above)
   - You mention supersaturation quite a bit but is it actually prognosed, or diagnosed based on a parametrization that includes updraft? And all diagnostic activation parameterizations depend on w (line 463). A problem is that it's not just vertical motion that drives supersaturation but the total effects of dynamics.
   - Model radiation: the model top is at 2 km. Does this mean you ignore the influence of the gases above the domain. If so, this is a serious omission. A column of atmosphere should be patched above for radiation calls so that the radiative effect of gases is included.
   - The low domain top might also explain why your modeled cloud deepens too much in the afternoon: If in fact there were upper-level clouds and the model doesn't see them then your cloud top cooling will be too strong and your cloud will deepen more than it should
   - Is hygroscopic growth included in the aerosol radiative effects and optical properties?
   - Typically, radiation is called much more often than 10 minutes (usually order 20 s). What effect is this having on simulations?
   - I was surprised that the aerosol model uses the $6^{th}$ moment as one of its moments. That's typically a choice for rain (radar reflectivity = $6^{th}$ moment)
   - Cloud void space is the 1-CF. Why not speak in terms of the familiar cloud fraction and make the reader's life easier?
   - As noted by a reviewer, the earlier part of the simulation is probably affected by spin-up. This is worth checking so that your discussion of the 0:00-04:00 UTC period is robust.
   - Bottom of page 11, other reasons include incorrect surface fluxes, and model weaknesses.
   - Caption Fig. 7h: why mention SWHR (no lines) at night
   - Lines 370-371: The increase in cooling with higher N is only true for clouds with LWP < 25 g/m2.
   - Line 595: references?

---

## Author Comment (AC1)

We very much appreciate the encouraging summaries alongside constructive and super detailed major, minor, and line-by-line comments on our paper from both reviewers. These comments (now has been appreciated in the Acknowledgments), have led to a significant improvement of our manuscript, reflected from many massively rewritten sectors and sentences with typo or grammatical corrections in the revised manuscript. We have certainly made many additional modifications to the text as well. The following are our point-by-point responses to the reviewers' comments (here the reviewers' comments are displayed first in bold *Italic* font).

*Reviewer #1 Dr. Mónica Zamora Zapata*
*Major comments*
*• There are multiple writing mistakes along the document, along with confusing statements. Please revise the manuscript carefully. Some suggestions are given below, but still, readability is essential for a paper.*
We admit that the previous manuscript could be better prepared. As indicated in our detailed responses below that we have tried our best to rewrite certain sentences or even sectors of the paper and correct typos to improve the readability of the manuscript.

*• Cloud break-up is used to refer to a state where the cloud deck has a low cloud fraction, but there is no clear definition of it nor a discussion about it. Some sentences talk about earlier or faster break-ups but I'd suggest to treat the concept carefully to avoid confusions, specially since coastal clouds go through cloud dissipation and the process is a complex one already.*
The reviewer's point is well received. We use the term break-up to describe the stage in cloud diurnal cycle when the dense stratus cloud layer starts to become cloud blocks separated by cloud-void spaces, though the cloud-void space would generally increase until dissipation, but break-up here is not equal to the dissipation. We admit that this needs to be made clearly in the manuscript. We have revised the paragraph in original 295 (and in other places) to clarify the difference between break up and dissipation or clear up: "At 06:00 UTC, cloud deck covers the entire domain as seen in both modeled result and in observations (note the distinct cloud rolls in model results). Between 10:00 and 13:00 UTC, the CPP in layers between mean CBH and CTH decreases from near 100% to 90%. Near the two averaged values, CPP decreases more to reach near 60% and 80% at CBH and CTH, respectively. This leads to a less inhomogeneous cloud deck confirmed by the LWP map and the observation of the sky camera at 12:00 UTC shown in the middle row of Figure 5. Indeed, more cloud-free pixels begin to appear between clouds and sunlight is seen through the cloud deck by the camera. Finally, the CPP continues to decrease until the end of the convection phase with a maximum barely reaching 80%, and a value around mean CBH and CTH as low as 20% and 40%, respectively. This demonstrates the break-up of the cloud deck during convection and the cloud thinning. The bottom panels of Figure 5 show clearly the dissipation of a large number of clouds alongside substantially thinning of the others at 16:00 UTC PM. The LWP map (Fig. 5b) shows numerous thin clouds corresponding to those seen by the camera of Savè".

*• The case studied was said to be a late dissipation one. They hypothesized that precipitation evaporation could explain this. No further analysis was done using LES data related to this issue, I think it should be diagnosed. I wonder if studying a more usual dissipation time case would give the same results. A more critical comment could be appreciated.*

The reviewer's point is well received. We have added the following sentence in Line 174 in the original manuscript that reads as: "Nevertheless, our focus of this study is on the diurnal cycle of LLSC as influenced by aerosols alongside planetary boundary layer dynamics rather than examining the above hypothesis appeared to be related to a process beyond the local scale. Therefore, our model setting is made to specifically eliminate the influence of mid-cloud layer for the purpose". Regarding the selected case, it is the best one we can have in terms of availability of observations that we can use to constraint the modeling and to make comparison of modeling results with observations".

***Minor comments***
*• I'd suggest to simplify the Meso-NH model description, focus on the setup and how the data was used to create initial and boundary conditions, and leave the model details on Appendix if desired. In my opinion, later experiments on aerosol contributions can also be described in the model section. This is personal preference, as I've seen both around.*
We have the same feeling about the model description. This is the reason that we have only included a few model components that are keys to our modeling effort.

*• Also related to readability, the results section can be summarized greatly, focusing on the big takeouts instead of carefully describing every value presented in the figures. While a description of the time evolution of the different vertical profiles seems fitting for this type of work, I'd suggest to be concise, highlight the most significant processes or values, since everything else is available in the plots.*
The point is well taken. The revised Results section now contains more comparisons to the previous findings. Certain parts of the discussions have also been made more concisely as suggested by the reviewer. Also, the Section 4 now opens with the following statements: "Previous studies have indicated that the life cycle of stratus or stratocumulus within planetary boundary layer depends on the subtle balance among several critical while interconnected forcings including surface heat fluxes, cloud top and base radiative profiles, and thus turbulent mixing (*e.g.*, Stevens *et al*., 2005; Dussen *et al*., 2014, Ghonima *et al*., 2016). Apparently, our simulation results of the REF case support previous findings particularly for cases over land with surface sensible heat playing a significant role. Nevertheless, the role of aerosols in such a life cycle have rarely explored in-depth. Given the critical role of aerosols in determining cloud macro- and microphysical features and thus radiation, this is a must-addressed issue to advance our understanding of the LLSC life cycle. A unique component of our study is the deployment of an interactive aerosol and atmospheric chemistry module in this observation-constrained modeling effort. In the following section we will discuss roles of aerosol variations in both number concentration and chemical composition in influencing the diurnal cycle of observed LLSCs".

*• Cloud formation is said to occur around hour 2, which is probably already part of the LES spin-up time. Could this be an issue?*
As mentioned in Line 246 in the original manuscript, the simulation starts from 11pm previous day to have a 1-hour spin-up before July 3. Hour 2 is thus already 3 hours later. Giving the stable and dense stratus before sun rise, we believe this setting has less influence on the diurnal cycle of modeled clouds after sun rise.

*• As results are presented, there is no critical comparison with the references given in the introduction for similar processes. It is important when summarizing to compare and also state what is novel.*
We appreciate this excellent point raised by both reviewers. This has been improved by adding comparisons to previous findings wherever applies.

*• Time-series of cloud fraction and liquid water path, both said to be available from observations and the LES runs, could complement the analysis greatly. Please include them in one of your figures, and strengthen the physical description of the PBL processes as needed.*
The cloud fraction was displayed in a different form in, e.g., Fig. 4. We realize that the term "cloud presence probability" might read odd to many readers without a clear description. This has been done by modifying then moving a sentence in original L289 to a better place for Fig. 4 discussion, original L278. It now reads as: "Note that to analyze the cloud cover profile over the domain, the Cloud Presence Probability (CPP) at each model layer, differing from cloud fraction that is often defined as a column metrics, is calculated as a percentage of all cloud pixels with a total condensed water mixing ratio exceeding $0.05$ g kg$^{-1}$ at the given model layer (Fig. 4)".

*• The cloud presence probability, CPP, is not carefully defined, so I don't understand how to interpret the values.*
Please see above response.

*Line by line comments*
*• L10 Is it necessary to remark that there is no precipitation like this in the abstract? Maybe combine with previous sentence "...effect, with all cases having negligible precipitation".*
We appreciate the suggestion. We would, however, keep the current sentences because 'all cases' might appear to be a stretching statement for DACCIWA, and the second sentence is to emphasize on the sensitivity to aerosol concentration.

*• L15 Why use cloud-void instead of cloud fraction or cloud cover? This is true for all the document.*
They are the two sides of the same coin. Though, cloud-void here would link closer to the cloud development associated with evaporation and thus offer a better direct reference to the process in discussion.

*• L21 What do you mean by stabilize? A more steady evolution?*
It means that heating near the cloud top would enhance thermodynamic stability of the cloud layer beneath and reduce the upward development. To avoid the confusion, we believe that "…lower the cloud top height" should be sufficient, therefore, "and stabilize the cloud layer" has been removed.

*• L23 Break-up faster means an earlier breakup time?*
Not necessarily, please consult the response to the corresponding major comment.
.
*• L64 Why would aerosol emissions impact the wind?*

There are several ways that aerosol direct or indirect effects could modify the energy budget and thus wind within the planetary boundary layer. Nevertheless, since this is not the main agenda of our research, "wind" has been removed.

*• L74 Are these 2 studies observational or model-based?*
The sentence has been modified to "Based on observations and parcel modeling, Taylor et al. (2019) and Denjjean ewt al. (2020a)…"

*• L80 It might be good to summarize what the direct, semi-direct, and indirect effects are, maybe in the second paragraph of the Introduction.*
A sentence of "This is because that aerosol can directly scatter or absorb solar radiation (direct effect or aerosol-radiation effect), or by serving as cloud nuclei, influence cloud microphysical structure and thus reflectance or lifetime (indirect aerosol effects or radiative effect of aerosol-cloud interaction plus cloud adjustment) (Boucher et al., 2013). The heating associated with aerosol absorption would be able to perturb atmospheric thermodynamic stability and thus dynamical processes as well (semi-direct effect) (Hansen et al., 1998). All these effects can modify the energy budget and thus the status of the planetary boundary layer where the stratiform clouds form."

*• L74 This paragraph has a lot of info but it's hard to follow as there is no clear story in it. The last sentence helps, but maybe it'd be good to have a similar introductory sentence and then highlight the knowledge gaps as info is presented. Just a suggestion. Another idea is to remove all unnecessary mentioned results if they won't be used for comparison or contrast.*
We agree with the reviewer that this paragraph is a bit too long. However, we believe that it serves a good purpose to describe major findings from the previous studies including aerosol sources and chemical compositions that benefit our study in many ways. And, as the reviewer appreciated that it has also indicated the shortcomings of the previous works particularly regarding aerosol-cloud interaction. Therefore, we decide to keep it here but with certain modifications, including the opening sentence now reads as: "Based on observations and parcel modeling, Taylor et al. (2019) and Denjean et al. (2020a) showed…".

*• Fig. 1 Maybe a proper reference is better than the link.*
It is from the publicly accessible image base; thus, the link is provided.

*• L98 What local scale and fine resolution are you talking about?*
We believe this sentence is a general statement suitable to be here. The model description coming later are sufficient for the details.

*• L116 Local time is also UTC for this location, right?*
"…(local time of Benin is UTC+1)" has been added.

*• L124 The info in this paragraph is a bit heterogeneous; some descriptions are very detailed and others not as much. Is the importance given to the details of the sky camera meaning that this data is more important than the rest? Also, a matter of preference, but it could be easier to follow with a consistent sentence structure like: first the instrument, then the data it generates (as done first with the ceilometer).*

We have removed certain details particularly for sky camara to balance the descriptions for different instruments.

*• L135 Aren't these fluxes included in the last sentence?*
The redundant "…as well as sensible and latent heat flux" has been removed.

*• L162 Turbulent mixing maintains cooling? You mean through downdrafts or are you referring to the cooling advection?*
The sentence has been modified as: "Turbulent mixing beneath the NLLJ alongside strong radiative cooling at the cloud top leads to the persistence…". The original sentence only described the sensible heat effect.

*• L168 What is the scenario C?*
Scenario C was summarized by Zouzoua et al. (2021) as indicated in the text and is actual elaborated in the following sentences.

*• L170 What does break-up mean here?*
Please see the response to the major comments and related modifications.

*• L192 Add reference instead of link.*
The link is removed. The reference provided in the opening paragraph of the sector should be sufficient.

*• L243 This resolution is not particularly high for stratocumulus LES*
We agree that the adopted vertical resolution is not the highest though quite typical. We have added the following text in discussion: "Note that previous studies regarding nocturnal stratus-stratocumulus suggested that a vertical resolution as fine as 5 meters near the cloud top would be necessary for reproducing the cloud top entrainment and thus cloud macrophysical structures (Stevens et al., 2005). Since the nocturnal-diurnal life cycle in our case involves a dynamically evolving cloud top (particularly in the daytime), it makes it difficult to prescribe a highlight zone for finer resolution. Our fast-testing results did not suggest an alarming difference between the run with 10 m and 5 m vertical resolution (not shown). Therefore, the current vertical resolution and the time step are selected to well cover all possible cloud tops during the simulation time and to provide the best economic computational performance for aerosol-cloud interaction with a fully coupled chemistry model".

*• L247 Are these values used to prescribed turbulence related to the local observations in any way? Is it okay for the profile to not change over time when modeling a coastal case?*
High pressure system and associated substance has been well documented.

*• L249 if the model was tuned for the site observations, it'd be good to include that in an appendix. Otherwise, what kind of parameters is this surface model taking in for this case?*
The Code and data availability has provided sufficient information for the data sources.

*• L251 Are these 2 s chosen for numerically advancing the LES? If so it's more than adequate for diurnal variations, and probably needed not for that reason but for keeping a good CFL number. It's not clear as is written.*
"…particularly involving aerosol and cloud microphysics" has been added.

*• L254 Do you mean using those observations as the initial condition? What is the nudging timescale for the tendency profiles?*
The words of "hourly radiosondes" have been added.

*• L258 How did you combine all that data to produce vertical profiles?*
This is quite a normal process for deriving profiles to drive LES runs. The quantities mentioned for the period beyond the availability of hourly radiosonde are the ones to better constrain the modeling. We have added sentences (in respond to the comment from another reviewer) to elaborate it: "Note that, despite these best possible efforts in configuring a set of observation-constrained tendency profiles to reproduce observed cloud field, it is difficult to eliminate the possibility that such profiles could reflect certain local thermodynamic effects however small they are. In practice, our principal is to make the profiles to be able to force the modeled clouds reproduce observed quantities of major features such as cloud top, base, LWP, surface incoming solar radiation in the REF case. This would serve the best purpose for us to address the major issue of this study, i.e., the role of different aerosol profiles in the diurnal cycle of modeled LLSCs".

*• Fig. 3 Improve the description: mass composition of what? Percentage text in a) is overlapping. Is the dot needed in the units of the b) y axis label (and all other labels)?*
The caption reads now as: "Aerosol chemical mass compositions (a) and size distribution fitted into 2 modes described in Table 1 (b) used in REF."

*• Table 1 I'd suggest putting the symbol for each parameter after mentioning them in the description.*
Done.

*• Fig. 4 and other figures are vertically shrinked. "Vertical colored bars" are actually dots. Do simulated values represent the horizontally averaged CBH? Why are they reported hourly when the LES has a smaller step size? What is the meaning of this probability?*
The figure has been adjusted to have a proper aspect ratio. CBH and CTH were both described with the term of "mean" in text and figure caption. Hourly display is due to the limit of output data (storage). Layer-defined Cloud Presence Probability has been better defined in the revised manuscript (see the response to a similar comment).

*• L273 Do you say "formation" because the initial state for the LES is cloud-free? If so, initial conditions should be stated carefully.*
Yes, it started from cloud free. This is a common approach. The initialization at 23:00 UTC in the previous day (thus with 1 hour spin up) was clearly described in the 3rd paragraph of 2.4.

*• L274 Is the mean CBH estimated as the horizontal average of points where the first cloudy grid point is located?*

Yes. We also added "domain" before mean in certain places.

*• L280 You can be more specific than "some period"*
It was indicated in the following words, "particularly late morning and afternoon".

*• L284 Could the difference also be due to prescribing a weak subsidence profile or due to enhancing entrainment by not having a very fine vertical resolution?*
"the vertical resolution of radar profiles" has been added.

*• L289 This aids the previous paragraph in saying that for 10:00-14:00 UTC, observed CBH is below the 0 level probability. I'd move this and start a new paragraph with the spatial results.*
*• L289 What does a CPP of 50% mean? Is it the geometric midpoint?*
For the above two comments: the opening sentence has been modified and moved to two paragraphs before. Yes.

*• L294 By visible camera do you mean sky images in the visible range?*
"sky camara (visible range)" has been added.

*• L295 "distinct cloud rolls" instead of "clear cloud rolls", so that clear is not confused with clear sky. It can also be noted that this feature is not observed in the sky image.*
Done as suggested.

*• L296 What is the CPP at the mid distance? At the mean in-cloud height? I still don't fully get the meaning of CPP values, so I don't understand what is useful of this description*
Revised to "in layers between mean CBH and CTH…".

*• L301 Rather than demonstrating the break-up, it evidences the already broken field. This comment is related to what is the definition of break-up.*
With the clarification of "break up" in place (see the relevant response previously), this sentence should be fine now. Though, "break-up" in the following sentence is inaccurate, it has been revised to "dissipation of a large number of clouds alongside substantially thinning of the others".

*• L303 Still, the camera at Sav`e shows a big portion of the sky completely clear. The LES does not reach that type of organization.*
This comparison means for qualitative not exactly quantitative purpose, and the sky camara's image and the model has different resolution as well, not to mention the tiny white blocks in camara's image are hard to identify.

*• L317 Do you mean that the difference is due to the tuning being done for more persistent cloud decks?*
As described in the newly added description of the model design, the reason is likely to be a mid-layer cloud way above the model top, which has been ignored on purposely.

*• L319 Why would that difference be reduced in the convective phase?*
The sentence has been removed because the result is insignificant.

• *L354 along what altitude?*
Added "above roughly 200 meters".

• *Fig. 7 θe has not been defined, and its units are just K, not ◦K. Put w s in equation mode. Why the discontinuous lines in f)? Why the choice of plotting T, RH, and θe instead of just θe and qt?*
$\Theta e$ was defined in figure caption as equivalent potential temperature, which is a commonly used metrics. "w_s" has been corrected, thanks.

• *L376 Does stable mean constant cloud thickness here? Be careful not to be confused with thermal stability.*
Yes, and, thanks for the reminder. With "cloud layers" here we assume it should be sufficiently clear.

• *L383 is this a fixed threshold to find the inversion height?*
No, it is apparently determined by the profile.

• *L416 Does more unstable mean signs of decoupling?*
Not necessarily.

• *L417 What does "end of cloud break-up" mean? That the clouds cleared up or that it's fully covered again?*
Revised to "coincides with the dissipation of the LLSCs and indicates".

• *Fig 9. Why report at 17 h and not 18 h to follow the 2 hours spacing? w s to formula. Why does TKE, RH and θe increase above the PBL at 17 h?*
Most simulations ended at 17h, only a few test runs went beyond. A that moment, clouds are almost all cleared and planetary boundary layer top leveraged. TKE is likely due to wind shear above. "w_s" corrected.

• *L435 What biases are you talking about?*
As discussed in the reference case. "even though …biases" has been revised to "despite certain biases".

• *L440 Do these cases represent extreme situations in the set of observations?*
As described here that these runs are based on actual observations. We assume true extreme situations are normally hard to capture in field campaign with limited time frame.

• *L456 What is cloud extent? Cloud thickness?*
Please consult with Fig. 4 and 5.

• *L464 What does "this way" mean? I'm guessing not by the model activation but the text is confusing.*
Revised to "under this condition".

*• L460 Here and in other places, comparing numbers in a more descriptive way can help the reader. For example, "NPOLc reaches half of the droplet concentration of the REF case (333 vs 653 droplets per cm3), with a slightly higher radius (6.45 vs 5.1 μm)."*
We have tried our best effort to make the comparison more readable.

*• Fig 10 Why are there 2 cloud layers at hour 6? It might be useful to also have a plot of the standard deviation of surface SW irradiance to accompany the last panel.*
Please note that these are layer-averaged quantities while cloud top might not be always flat. Nevertheless, based on the values of the quantities, the second cloud layer (if we understand the comment correctly) does not look physically stable or even sound.

*• L473 I'd suggest to check if the difference in LWC is related to changes in cloud cover. Also, since the changes in CTH and CBH are equivalent to just a few grid points, dive into that discussion as well, what was the expected outcome? Are there both positive and negative cloud thickness feedbacks that may be canceling each other?*
It is hardly to understand cloud cover change would lead to a different liquid water content easily – the latter is defined by profile for a relatively well-mixed planetary boundary layer in our case. As what we see from the modeled results (alongside those of many others), condensation could be another reason to cause the difference since a sufficiently higher cloud droplet number concentration generally lead to a higher liquid water content.

*• Fig 11 It is hard to distinguish the clear portions in the last panel. Maybe you can set the zero values as NaN for plotting them in white, and mention that in the caption. I don't know if it's related to the wind at the time but POL seems to have less elongated structures at 14 UTC. What is ADEON?*
The difference is visible from the figure though not as large as in REF vs. CLEAN. This figure is actually a result from numerous testings with different color schemes and skills including NaN or alike. ADEON is not necessary to be here and has been removed, thanks!

*• L491 When saying "major" reason, what is the other reason why the trend would be different from what you expected?*
The normally expected outcome is described clearly in the following sentences.

*• L492,L500 The solar zenith angle should be lower near noon, right? Do you mean solar elevation angle?*
Thanks, corrected.

*• L498 I don't understand this sentence well. When you say "direct solar flux is relatively low" you don't mean the time of the day, right? Then you talk about cloud reflectance, which I'm guessing you interpret through LWP or LWC, saying it is a secondary factor. Does this mean that for clouds without full cloud cover (which would be the primary factor), then changes in reflectivity are also promoting the unexpected result for the POL case?*
Generally, yes to the first part of question except for that short-wave cloud reflectivity is related to the cloud droplet number concentration rather than total condensed water. To avoid confusion, "When direct…" has been revised to "Thus, before noontime cloud reflectivity…". The second part, no, the primary factor during this stage is the solar radiation reaching the ground directly by

passing through cloud-void space. Otherwise, more reflective cloud, as expected in the so-called classic Twomey effect, would lead to a lower amount of solar radiation reaching the ground.

*• L500 By "this", do you mean the reduced PBL heating due to a higher reflected SW at cloud top?*
"This" is revised to "The lower SWRADSURF in CLEAN".

*• L504 Though the images do suggest that some layers may dissipate earlier, it'd be better to include time-series plots of LWP and cloud fraction, maybe as panels b and c in Fig. 4.*
The point is well received. However, since the time series of LWP and cloud fraction (though layer-defined) have already provided in Fig. 4 and 5, and additional information of microphysical features in Fig. 10, we feel that adding one more panel in Fig. 4 might not serve the best of purpose.

*• L505 The last sentence is confusing. It seems to relate a low number of variable pixels with variations in surface SW irradiance.*
It has been removed.

*• Table 3 Is this LWP or LWC (units of LWP are typically g/m2)? Why don't you put the % symbol instead of writing percentage again? Last column title shouldn't be PRP, right?*
It is LWP, g/kg m is commonly used for lower atmosphere without involving air density. Percentage in Table caption is necessary, this would save the space in the Table.

*• L511 Do you think that a different type of variation could cause the more expected result?*
Thanks for the thought but this is apparently an open question now, perhaps worthy another effort.

*• L523 You should explain the REF NOBC case here (I'm guessing it has no BC, but were all the other aerosols kept?)*
This is indeed a sentence skipped being removal before submission. It has been removed except for the first sentence, revised as: "Apparently, BC is the major species behind the semi-direct effect in our case study."

*• L534 You could also include this mentioned difference in liquid water content in Fig. 12, because it's not shown.*
Please see the previous relevant response.

*• Fig. 12 It might be useful to include cloud thickness in panel a too. Why are the first hours skipped? If break-up time is important, it could be marked in these plots.*
Good point, though we already have too many figures and panels. Stable cloud formed at hour 2.

*• L541 I don't completely understand the last sentences. If the CTH is kept nearly constant, dissipation could still occur due to other factors, are all of them unchanged? This could be diagnosed using your difference approach on a sort of budget terms (see van der Dussen 10.1175/JAS-D-13-0114.1 and Ghonima 10.1175/JAS-D-15-0228.1 works on LWP and cloud thickness budget equations, I'm not sure if there's work relating them to aerosol effects).*

We assume the comment is on the effect of BC in lowering the cloud top. Please note that we have simulated rather a long time period from nightly stable to daily convective planetary boundary layer. During the latter stage, cloud top has always been evolving as shown in figures. The point is well received. Findings from the two works suggested by the reviewer alongside others have been discussed when necessary. An example is the newly added opening paragraph of Section 4 as indicated in a previous response.

*• L550 This being said, maybe the analysis could benefit from comparing not only the domain averaged SW fluxes but by separating the domain in cloud-void and cloudy portions, in order to quantify how much the low cloud fraction effect weights.*
Thanks for the thought. This is a largely Ph.D. thesis work. There are more questions than answer as the reviewer perhaps understand well. They could all be addressed in future efforts.

*• L535 This decrease in TKE is very interesting. I'd interpret it as limiting entrainment.*
Added, thanks!

*• L554 How do you know that clouds break up slower? If you mean a state with greater or lower cloud cover, I think that's different from a break-up speed. Still, a cloud cover vs time plot could hint towards that.*
*• L555 Note that clouds are also larger for the ADEOFF cases at 16 UTC, this is relevant for cloud organization and solar variability.*
Response to above two comments: "Fig. A3" has been added as the reference.

*• L559 This is a bit confusing. Are these the effect of having BC or of not having it?*
It is referred to a realistic situation, so BC is certainly included and thus the root cause.

*Typos/Writing suggestions*
We very much appreciate a massive and time-consuming effort of this reviewer in providing a detailed list of suggestions to improve the readability of our manuscript. All suggested changes have been done. To save the space, the full list of responding changes is not shown here.

---

## Author Comment (AC2)

We very much appreciate the encouraging summaries alongside constructive and super detailed major, minor, and line-by-line comments on our paper from both reviewers. These comments (now has been appreciated in the Acknowledgments), have led to a significant improvement of our manuscript, reflected from many massively rewritten sectors and sentences with typo or grammatical corrections in the revised manuscript. We have certainly made many additional modifications to the text as well. The following are our point-by-point responses to the reviewers' comments (here the reviewers' comments are displayed first in bold *Italic* font).

**Reviewer #2**
*MAJOR issues:*
*-place results in context with previous studies, both when describing the reference case as well as the aerosol sensitivity studies.*
This has been improved. Relevant previous findings including those from three additional references suggested by both reviewers have been discussed accordingly in the result sections. As an example, the Section 4 now opens with the following introduction:
"Previous studies have indicated that the life cycle of stratus or stratocumulus within planetary boundary layer depends on the subtle balance among several critical while interconnected forcings including surface heat fluxes, cloud top and base radiative profiles, and thus turbulent mixing (*e.g.*, Stevens *et al.*, 2005; Dussen *et al.*, 2014, Ghonima *et al.*, 2016). Apparently, our simulation results of the REF case support previous findings particularly for cases over land with surface sensible heat playing a significant role. Nevertheless, the role of aerosols in such a life cycle have rarely explored in-depth. Given the critical role of aerosols in determining cloud macro- and microphysical features and thus radiation, this is a must-addressed issue to advance our understanding of the LLSC life cycle. A unique component of our study is the deployment of an interactive aerosol and atmospheric chemistry module in this observation-constrained modeling effort. In the following section we will discuss roles of aerosol variations in both number concentration and chemical composition in influencing the diurnal cycle of observed LLSCs".

*-Language-related issues, like grammatical errors and long and complex sentences make following the manuscript a difficult task. Please correct all grammatical mistakes (only some examples given along the line by line comments) and keep sentences short for the shake of clarity.*
We truly appreciate these massive line-to-line comments from both reviewers. Improving the readability of our manuscript has been done with our best effort as partially demonstrated from our responses.

*MINOR issues:*
*-Please clarify how exactly the tendency profiles are obtained from the radiosoundings. As it is now, I understand that the tendency applied to the LES domain over each hour is equivalent to the difference in T (and q) between two consecutive radiosoundings divided by the time passed between these. My current understanding is that such tendencies are used as proxies for large scale advection of moisture and temperature. If this is the case, I am afraid such tendencies may include not only the evolution of temperature and humidity due to large scale advection, but also the tendencies due to local thermodynamic effects such as radiative cloud top cooling or warming/moistening (after sunrise) due to surface fluxes.*

The reviewer's point is well received. Indeed, the procedure of creating the tendency profiles is largely as the reviewer mentioned above, and we agree that there is a chance that the local thermodynamic effect could affect could be included in the profiles. We have tried our best to minimize such an artifact by forcing the modeled clouds to follow as close as possible the observed quantities such as cloud top and base as well as surface incoming solar radiation. We have added the following sentences in the revised manuscript:

"Note that, despite these best possible efforts in configuring a set of observation-constrained tendency profiles to reproduce observed cloud field, it is difficult to eliminate the possibility that such profiles could reflect certain local thermodynamic effects however small they are. In practice, our principal is to make the profiles to be able to force the modeled clouds reproduce observed quantities of major features such as cloud top, base, LWP, surface incoming solar radiation, among others, in the REF case. This would serve the best purpose for us to address the major issue of this study, i.e., the role of different aerosol profiles in the diurnal cycle of modeled LLSCs."

***-Section 3 goes through the results of the REF simulation in high detail. Readers would find it easier to understand, however, if instead of a description of each result, a more concise section with the most relevant results is presented. This would also allow, as suggested in the fist Major issue, some room to link the relevant results to previous studies.***
We appreciate the point of the reviewer. We have linked certain discussions with previous findings. Certain parts of the discussions have also been made more concisely.

***Line by line comments:***
***L44: Please introduce briefly the direct, semi-direct and indirect effects of aerosols, given they are recurrently mentioned along the manuscript.***
The following has been added after the opening sentence of the second paragraph of Introduction: "This is because that aerosol can directly scatter or absorb solar radiation (direct effect or aerosol-radiation effect), or by serving as cloud nuclei, influence cloud microphysical structure and thus reflectance or lifetime (indirect aerosol effects or radiative effect of aerosol-cloud interaction plus cloud adjustment) (Boucher et al., 2013). The heating associated with aerosol absorption would be able to perturb atmospheric thermodynamic stability and thus dynamical processes as well (semi-direct effect) (Hansen et al., 1998). All these effects can modify the energy budget and thus the status of the planetary boundary layer where the stratiform clouds form".

***L67: remove dot.***
Done.

***L87: what effect?***
The sentence has been modified as: "…that sedimentation of cloud droplets, determined by droplet size, could affect liquid water path by removing droplets from the entrainment zone, or by lowering the cloud base and creating more heterogeneous cloud structure".

***L135-136: I see surface fluxes first, and sensible and latent heat fluxes later on. If it is referring to same measurements please delete one of the references. If they are different measurements, please clarify.***

The redundant "…as well as sensible and latent heat flux" has been removed.

***L141 analyzer instead of analyzed***
"analyzed" has been removed.

***L146; What is the approx spatiotemporal resolution of these measurements?***
These are airborne measurements so the resolutions are only relative to the path, and these should be indicated in the cited reference. The sentence has been revised to: "Meteorological variables such as temperature, humidity, pressure, and wind speed and direction were also measured by a suite of airborne instruments."

***L164: On top of the surface heating, I would expect also a weaker cloud top cooling due to solar radiation being absorbed at cloud top. If this is the case, please mention it. And if it is not, please explain why.***
"alongside a weak radiative cooling at cloud top (e.g., Ghonima et al., 2016)" has been added, thanks for the reminder!

***L186 remove comma after scales***
Done.

***L187 transport***
Done.

***L190 centered***
Done.

***L204: remove 'completing LIMA'.***
Done.

***L242: Stratus clouds are known to be very sensitive to the vertical resolution near cloud top (Stevens et al 2005). It would interesting to learn a bit more on the sensitivity of this case to the vertical level spacing (if previous numerical experiments with different vertical spacing were performed), and why 10m was decided eventually as the vertical spacing for the lower part of the domain.***
Thanks for the excellent point. We indeed tested using various vertical and horizontal resolutions in early fast runs. The selection of 10-m is based on the performance and most importantly, the diurnal convective PBL that might be difficult to use a generalized highlight zone for finer resolution. Nevertheless, this is an important point to indicate, the following sentences have been added in the revised manuscript:
"Note that previous studies regarding nocturnal stratus-stratocumulus suggested that a vertical resolution as fine as 5 meters near the cloud top would be necessary for reproducing the cloud top entrainment and thus cloud macrophysical structures (Stevens et al., 2005). Since the nocturnal-diurnal life cycle in our case involves a dynamically evolving cloud top (particularly drastically in the daytime), it makes it difficult to prescribe a highlight zone for finer resolution. Our fast-testing results did not suggest an alarming difference between the run with 10 m and 5 m vertical resolution (not shown). Therefore, the current vertical resolution and the time step are

selected to well cover all possible cloud top during the simulation time and to provide the best economic computational performance for aerosol-cloud interaction with a fully coupled chemistry model".

*L255: Please explain further what is done regarding the tendencies for horizontal wind and the presence or not of a Nocturnal Low Level jet (since it was mentioned in L150 as being closely related to cloud formation).*
"and horizontal wind" has been added.

*-L256 Linked to minor issue n1. It would also be of interest to show the temperature and humidity profile obtained by the radiosoundings in (at least some) of the profiles in figures 7, 8, and 9.*
The point is well received. The figures are already quite messy, thus we are reluctant to add any more profiles to them. Fortunately, DACCIWA measurement data are publicly accessible, so the reader could obtain these profiles.

*Fig4. Adding the 4 phases introduced earlier and described below (stable, jet, stratus, convective) below the time axis would help the reader to locate the phases in this specific case and come back to it when needed along the manuscript.*
*L277: Please add an indicative UTC time for the onset of the convective phase to better guide the reader*
Done for adding the stages in Fig. 4 so the UTC time can be read directly from the figure.

*L318 'But, as the LLSCs…' It is difficult to understand the meaning of this sentence. Please rephrase.*
The sentence has been removed.

*L319 Any thoughts as to why the difference is reduced during the convective phase?*
The sentence has been removed since the difference is rather small.

*L372: I would substitute 'at the cloud' top by 'above the cloud top'. In fact, the strong longwave emission is a source for turbulence in the cloud layer as parcels near cloud top cool and sink along the cloud layer.*
The sentence has been modified to "above the cloud top, deepening the temperature inversion."

*L376: I am confused as the cloud layer is now called to be 'very stable', while one line above it was said to be well mixed.*
Agree that the sentence appears to be inaccurate, removed.

*L381 I am again confused by the use of 'stable' when the plot shows almost near-constant equivalent potential temperature.*
"stable" is revised to "well-mixed".

*L415: I find the concept of breakup confusing. In this line the breakup is said to happen at 16 00, but the lower Figures in Fig 5 at 16 00 suggest that, if the breakup is defined as the first*

*moment with LWP=0 somewhere in the domain, such breakup happenned earlier (even at 12 00, one could argue looking at FIgs 5 c,d).*

The paragraph discussing Fig. 5 (starts at original L296) has been revised to give a better description and to distinct break up from dissipation. This revised one reads as:

"At 06:00 UTC, cloud deck covers the entire domain as seen in both modeled result and in observations (note the distinct cloud rolls in model results). Between 10:00 and 13:00 UTC, the CPP in layers between mean CBH and CTH decreases from near 100% to 90%. Near the two averaged values, CPP decreases more to reach near 60% and 80% at CBH and CTH, respectively. This leads to a less inhomogeneous cloud deck confirmed by the LWP map and the observation of the sky camera at 12:00 UTC shown in the middle row of Figure 5. Indeed, more cloud-free pixels begin to appear between clouds and sunlight is seen through the cloud deck by the camera. Finally, the CPP continues to decrease until the end of the convection phase with a maximum barely reaching 80%, and a value around mean CBH and CTH as low as 20% and 40%, respectively. This demonstrates the break-up of the cloud deck during convection and the cloud thinning. The bottom panels of Figure 5 show clearly the dissipation of a large number of clouds alongside substantially thinning of the others at 16:00 UTC PM. The LWP map (Fig. 5b) shows numerous thin clouds corresponding to those seen by the camera of Savè".

***L401 It is difficult to understand this sentence, please rephrase.***

It has been revised to: "However, the turbulent kinetic energy increases to 0.1 $m^2 \, s^{-2}$ throughout the vertical layer from 50 meter above the ground to a level just below the cloud top. This enhancement of turbulence is expected to increase entrainment entering the cloud from above as well".

***L417: Now 16 00 seems to be the end of the breakup. Please clearly state how the breakup is defined and keep it consistent across the manuscript.***

The last sentence has been revised to "This increase coincides the dissipation of the LLSCs and indicates the arrival of the marine inflow".

***L420: I am confused as to at what altitude I should look for a 1.25 m2 s-2 TKE in in Fig 9. Please clarify.***

It should be "0.25", corrected.

***L421. Also in L387 vertical windspeed was assumed to be the driver fro TKE changes. I'd suggest therefore adding the profiles of windspeed, even in an appendix given the already full panel in Figures 7, 8, and 9.***

We very much appreciate the suggestion. However, as the reviewer knows that the small quantity of vertical velocity might not be a pleasure to read.

***L447: dividing, not deriving.***

Done.

***L465: It is very challenging to understand this sentence. Please rephrase and divide into shorter and clearer sentences.***

Revised to "This trend is reversed at 06:00 UTC when the droplets number concentration and radius are equal to 1208 *droplets cm*$^{-3}$ and 6.43 *μm* for POL, 1305 *droplets cm*$^{-3}$ and 6.12 *μm* for REF, respectively".

***L494: I don't think 'inverse layer' is the right term.***
Revised to "temperature inversion zone".

***L495 increase instead of exceed***
Revised to "…evaporation rate of droplets would exceed that in CLEAN case".

***L505 I cannot follow the last part of the sentence.***
It has been removed.

***L511 The explanation is interesting. To validate such hypothesis, it would not take too much effort to compute some metric for spatial SWRADSURF variability. If it turns out to be larger in POL and REF than in CLEAN, then the hypothesis of cloud-holes increasing the shortwave radiation reaching the surface would be reinforced. Further analysis of the cloud layer could also help, since all variables are present in a LES simulation.***
We appreciate the excellent suggestion. As the reviewer perhaps understands, this paper is a part of a Ph.D. thesis study that, as always, raises more questions than answers perhaps, and does leave quite a space to explore in future.

***L529 I would not call dispersion to cloud top and cloud base.***
Thanks! It has been revised as "cloud top and base".

***L549: I dont see how Fig12c contributes to the sentence. Please clarify.***
Revised, "Fig. 13c and A3".

---

## Author Comment (AC3)

We very much appreciate the comments from the handling editor Dr. Graham Feingold on our paper. The following are our point-by-point responses to these comments (the Editor's comments are displayed first in bold *Italic* font).

***In addition to the reviewers' comments, I have a number of important comments that I would like you to take into account before resubmission:***
***1) To begin with, the quality of the writing is not to up to par. I am sympathetic to the fact that the authors are probably not naive English speakers but we cannot underestimate the importance of clearly written and sharp text that communicates the ideas effectively. Unfortunately, much work needs to be done in this regard.***
We agree that the manuscript could be better prepared. The entire manuscript has now been largely rewritten. The revised manuscript should reflect our best effort in making a significant improvement of the paper's readability.

***2) On a scientific note, the authors do not seem to have strong familiarity with the literature on aerosol-cloud-radiation interactions and thus, while the results seem robust, some of the explanations are well-known and could be stated much more simply, with appropriate references. In many places this leaves the incorrect impression that the authors have discovered something new.***
As a general response to the editor's above comment, we have modified discussions in several sections of the paper, primarily to emphasize on the outcomes of our study that are different from previous analyses after introducing aerosol's role into consideration. For example, new discussions (and new figures) regarding LPW and CF response to both aerosol abundance and chemical composition have been added. We appreciate specifically the references suggested by the editor, and we have cited most of them to indicate the consistency as well as difference between results of ours and others.

***Examples:***
***- Liquid water path (LWP) and cloud fraction (CF) adjustments***
***The notion that the Twomey effect is the dominant one has long been shown to be inadequate, especially given the much stronger control of N on LWP (2.5 x more important in a relative sense) and the dominance of CF, about which much less is known regarding aerosol effects. There is a very large body of literature on this topic and in various places, the text comes across as naive (e.g., bottom of page 24, and bottom of page 25).***
We believe that what we expressed in the text, e.g., in the bottom of Page 24 alongside other places, is basically the same message as the above comment, i.e., the Twomey effect is not always the dominant factor in controlling the radiative effect of stratiform clouds particularly regarding the incoming solar radiation at the ground. To better understand the issue, the responses of LWP and cloud fraction to aerosol variation need to be considered. As the editor perhaps also agree that our effort is specifically to examine the role of aerosols on this regard comparing to many other works in the large body of literature, especially by taking both aerosol size distribution and chemical composition into consideration, aside from the key microphysical processes interacting with dynamics that has caused such an outcome. We certainly should express our points more clearly. Thus, we have made modifications in several related discussions. A new Fig. 12 and Fig.A4 have been added to show the timely variations of LWP

and CF in various runs. These newly added analyses indeed brought quite many revisions of Abstract, Conclusion, and main discussions. See the following response in more specific.

*- LWP adjustments are usually negative in stratiform clouds*
*(https://doi.org/10.1175/1520-0469(2003)060<0262:TCALWT>2.0.CO;2*
*https://doi.org/10.5194/acp-19-5331-2019, doi:10.1029/2006GL027648*
*This makes it a central variable for aerosol-cloud-radiation interactions, and yet other than the Table values, it's hard to get a good picture of LWP evolution and how N might be affecting LWP. The same is true for CF: the smaller the cloud fraction the smaller the radiative effect of the clouds, and the less leverage there is for aerosol effects on clouds. Aerosol effects on CF are less well quantified but this might be where some extra work gives you an opportunity to say something new.*

The editor's point is well received. As mentioned in the response to the previous comment, we have added new figures (new Fig. 12 and Fig. A4) to show the time evolutions of LWP and CF in different runs. New and revised discussions can be seen from Abstract, Section 4, and Conclusion, among others. In brief, our result of LWP response (negative) to CDNC is in an agreement with the previous studies (note that our configurations for aerosol-CDNC relation brought in size distribution alongside chemical composition). We agree that the CF-aerosol has rarely been addressed in-depth, and with the unique modeling configurations we should be able to obtain some new knowledge. We now show that CF-CDNC relation varies in different stages, for most of the convection stage before massive cloud break-up occurs, they are inversely correlated. In addition, semi-direct effect causes a lower CF in the same period.

Here is one example of the revisions (newly added in Section) 4.2): "Looking into timely varying metrics of LWP in various run, we find that in general, LWP is inversely promotional to CDNC, as LWP in POL < LWP in REF < LWP in CLEAN, and this is applied to different metrics of LWP (Fig. 12, Table 3). However, in comparison, the peak LWP varies less significantly in CLEAN case, while peak LWPs in two other runs decrease with domain averaged quantities in convection stage. There were different opinions regarding why such an inverse relation between LWP and CDNC (or aerosol number concentration in works with simple aerosol-cloud model) (*e.g.*, Ackerman *et al*., 2004; Bretherton *et al*., 2007). In our analysis, the difference in turbulent mixing driven by the surface radiative heating, as influenced by different microphysical features in various cases, seems having played a critical role. The situation of cloud fraction (CF) is somewhat more complicated. As shown in Table 3 and Fig. A4, CF relation with CDNC varies in different stages. An inverse relation between CF and CDNC generally stands in the earlier and later period of convection stage, in the mid-convection stage (13:00-15:00 UTC), the above relation would reverse, not to mention that the difference in cloud extent among different runs as discussed previously".

In 4.3, a newly added paragraph regarding semi-direct effect reads as: "The impact of the semi-direct effect on other critical macrophysical features such as cloud fraction and LWP can be also seen from the model results. For instance, LWPs are clearly lower in the AODON runs of the two polluted cases (REF and POL) (Fig. 12). In addition, an increase of cloud fraction due to the semi-direct effect can be seen throughout the convection stage until 15:00 UTC when massive cloud break-up occurs (Fig. A4). All these imply a critical role of semi-direct effect in cloud radiation".

*- On page 21, the discussion of the microphysical responses is long and not very informative because much is already anticipated. Is the POLLUTED case even needed given that REF is so polluted already and that at some point updraft/supersaturation production cannot activate any more aerosol?*

The indicated discussion has been rewritten to simplify description of certain expected results while emphasize on some special cases. By the way, in the discussion regarding the role of different evaporation rate due to aerosol size distribution, Wang et al. (2003).

Regarding whether the POL is even needed since REF is already polluted, we believe the answer is yes. This is because that size distribution and chemical compositions of these two observed profiles are different despite that they have the similar peak number concentrations. Such a difference could cause different semi-direct effect aside from activation, as suggested by the results. We have revised 4.1 to emphasize the differences between POL and REF aerosol profiles. In addition, discussions have also been revised to indicate different modeled outcomes associated with the above difference.

One example of newly revised discussions reads as: "At the cloud formation (02:00 UTC), despite having similar liquid water content (LWC) around 0.35 $g\ m^{-3}$ at 250 m in both cases, $N_c^{POL}$ reaches 333 $droplets\ cm^{-3}$ and $r_c^{POL}$ 6.45 $\mu m$ instead of 653 $droplets\ cm^{-3}$ and 5.1 $\mu m$ for REF case, indicating a result of differences mainly in the Mode 2 aerosol numbers between the two scenarios (at 02:00 UTC the updraft near cloud base is rather weak at less than 0.30 $m\ s^{-1}$ in both cases). This trend is reversed at 06:00 UTC when the CDNC and radius are equal to 1208 $droplets\ cm^{-3}$ and 6.43 $\mu m$ in POL, and 1305 $droplets\ cm^{-3}$ and 6.12 $\mu m$ in REF, respectively. After 08 UTC and until the cloud break up, $N_c^{POL}$ is superior to $N_c^{REF}$ by reaching a maximum difference of 1425 $droplets\ cm^{-3}$ at 14:00 UTC. Their respective radii are 4.42 $\mu m$ and 5.18 $\mu m$ while the liquid water content profiles are quite the same as near 0.47 $g\ m^{-3}$ at 750 m. The difference between POL and REF in CDNC after sunrise suggests that the activation favors the POL profile with higher sulfate content when updraft is strengthened".

*- Effect of drizzle: it has been proposed that weak drizzle can stabilize clouds (by preventing deepening https://doi.org/10.1175/1520-0469(1998)055<3616:LESOSP>2.0.CO;2) and if drizzle evaporates just below cloud base, can strengthen turbulence by destabilizing the BL (doi:10.1029/2001JD001502) When discussing drizzle, please engage in these ideas and see if they are relevant to your analysis (e.g., CLEAN, Fig. 10). In Fig. 10, a TKE profile would help to show whether weak drizzle just below cloud base might be enhancing cloud turbulence/deepening. You could show divergence of the modeled drizzle flux to get a sense of evaporative cooling below cloud base.*

We appreciate the excellent point regarding drizzle. We did not expand our discussion on this aspect is primarily due to the fact that drizzle was believed to be rare for this case based on observations. From our modeling results, it can be clearly seen from the profiles that the cloud droplet size is rather small (sub-10 micron; Fig. 7, 8, 9, & 10), and the formation of drizzle sized particles were rare even for the CLEAN case despite a "drizzle-sensitive" scheme of Khairoutdinov and Kogan was specifically introduced in our modeling. The point is still valid, we have added the following sentence in discussion: "Despite the relatively larger size of the droplets in CLEAN comparing to the cases of POL and REF, there is no clear sign of massive formation of drizzles even in the convection stage (Fig. 10). Nevertheless, sedimentation thus evaporation of larger droplets from cloud base could likely create a thermodynamic perturbation

(*e.g.*, Stevens *et al*., 1998; Jiang *et al*., 2002), though the quantity of such a perturbation seems rather small here".

*- Where does the absorbing aerosol reside? This makes a significant difference to the dynamical response (e.g. doi: 10.1256/qj.03.61, doi:10.1029/2005JD006138, 10.1002/2015GL066544). And please convey the essence of knowledge already known from these papers, rather than simply providing lists of references. The current version of the text is not careful about using those references to provide context.*

Aerosols are relatively well mixed within the PBL (the initial profile and the periodic lateral boundary condition) (e.g., Fig. 7, 8 & 9). We did not apply any specific isolated BC layer as in the referred idealized studies. We have added a few more sentences in the revised manuscript in Introduction and in discussions. For example, the new sentence in Introduction reads as: "Such a semi-direct effect can be positive or negative depending on the relative distribution of the aerosol with respect to clouds (*e.g.*, Johnson *et al*., 2004; Feingold *et al*., 2005)". Also in 4.3, "Note that our modeling configurations are based on aerosol profiles that are relatively well-mixed throughout the PBL then with concentration gradually decreasing along altitude above PBL. Certain previous sensitivity experiments suggested that the location of BC layer within or above PBL could have different impacts on the development of convection, entrainment, and thus life cycle of the low clouds within PBL (*e.g.*, Johnson *et al*., 2004; Feingold *et al*., 2005)".

*3) Missing information/other comments:*
*- The cloud radar is mentioned but we aren't told its wavelength, which makes it hard to interpret what it sees. (See drizzle discussion above)*

We have added information of the Ka band mobile, dual-polarization Doppler radar (8.5 mm, 35.5 MHz) in the revised text.

*- You mention supersaturation quite a bit but is it actually prognosed, or diagnosed based on a parametrization that includes updraft? And all diagnostic activation parameterizations depend on w (line 463). A problem is that it's not just vertical motion that drives supersaturation but the total effects of dynamics.*

The activation is calculated using Abdul-Razzak and Ghan scheme (2004) as described in the text, and thus supersaturation is obtained from diagnostic method with several corrections mainly at cloud top. We have made these clearer in the revised manuscript, in the description of LIMA and ORILAM. We certainly agree that considering 3D air motion could in theory lead to a more accurate estimation, though the difference might not be so significant comparing to the outcome derived by vertical-dominated diagnostics, because the large temperature gradient in vertical direction and the maximum activation near the cloud base. Based on our knowledge, the current effort is still on making a better correction at cloud top.

*- Model radiation: the model top is at 2 km. Does this mean you ignore the influence of the gases above the domain. If so, this is a serious omission. A column of atmosphere should be patched above for radiation calls so that the radiative effect of gases is included.*

Meso-NH model is a community model just like WRF and developed for various applications, not all of them would have a domain to cover the whole atmosphere. The radiation above the model domain, as in any other such modeling application, is calculated based on prescribed (e.g., climatological) profiles. To avoid unnecessary misunderstanding, we have added in the revised

text that: "Note that the radiation module still proceeds calculations above the 2 km using prescribed profiles".

***- The low domain top might also explain why your modeled cloud deepens too much in the afternoon: If in fact there were upper-level clouds and the model doesn't see them then your cloud top cooling will be too strong and your cloud will deepen more than it should***
Comparing to many other similar modeling efforts, our domain is not necessarily too low considering the vertical extent of the cloud layer. Certainly, a mid-cloud layer with drizzle could lead to a faster break up as hypothesized (cited in the paper), or without this mid-cloud the modeled cloud would actually last longer than observed one did. Nevertheless, this is the reason why we decided to design a configuration to avoid this issue, rather than focusing on the aerosol impacts within lower atmosphere. We have added a statement regarding this in responding to the reviewer #1 comments, as "Nevertheless, our focus of this study is on the diurnal cycle of LLSC as influenced by aerosols alongside planetary boundary layer dynamics rather than examining the above hypothesis, which appeared to be related to a process beyond the local scale. Therefore, our model setting is made to specifically eliminate the influence of mid-cloud layer for the purpose as described later".

***- Is hygroscopic growth included in the aerosol radiative effects and optical properties?***
The aerosol code ORILAM has a gas-particle equilibrium (EQSAM for inorganics and MPMPO for organics) that allows the model to calculate the water content of the aerosol. The solver will then combine the moment 0 (integrated number) and 3 (integrated new volume which integrates the hygroscopic growth) to calculate the new dimensional distribution (Tulet et al., 2005, 2006; both were cited in the manuscript). The parameterization of aerosol optical properties introduced in Meso-NH (Aouizerats et al., 2010; cited in the manuscript) allows the calculation to be processed at each time step, with a refractive index corresponding to the chemical composition of aerosol particle (including the amount of water) according to the Maxwell-Garnett equation (Maxwell-Garnett, 1904) as defined in Tombette et al. (2008). This approach considers that the aerosol is composed of an inclusion and an extrusion. The inclusion is composed of the primary and solid parts of the aerosol, while the extrusion is composed of the secondary and liquid parts of the aerosol. These are largely described in the manuscript already. We have reorganized the contents with certain added information regarding these aspects in the revised manuscript, in the paragraph of ORILAM description.

***- Typically, radiation is called much more often than 10 minutes (usually order 20 s). What effect is this having on simulations?***
It has been tested with values as short as 30 seconds, the step is selected as an optimized solution based on the modeled outcomes and also overall computation burden.

***- I was surprised that the aerosol model uses the 6th moment as one of its moments. That's typically a choice for rain (radar reflectivity = 6th moment)***
The ORILAM aerosol model adopts a 3-moment approach to close the log-normal distribution defined by a median radius, an integrated number, and a dispersion. The choice of moment 6 is numerical, it allows one to calculate the coagulation coefficients explicitly and to facilitate the integration of the aerosol solver. All these were described in detail in the two cited Tulet et al.

papers. We have also added a note about the selection of the 6th moment in the ORILAM description paragraph.

*- Cloud void space is the 1-CF. Why not speak in terms of the familiar cloud fraction and make the reader's life easier?*
The point is well received. We have now used the term of cloud fraction as much as possible in the discussions unless when cloud void space appears to be more natural in expression (positive, negative or increase and decrease). By the way, the axis label of "Cloud Presentation Probability" in Fig. 4 and A2 have been replaced by Cloud Fraction with a notation in the caption and associated text to mark the layer-defined nature of this quantity.

*- As noted by a reviewer, the earlier part of the simulation is probably affected by spin-up. This is worth checking so that your discussion of the 0:00-04:00 UTC period is robust.*
This has been checked through testing runs in the early stage of the modeling. The simulation started at 23 UTC of the previous day, though the cloud formed around 2 UTC based on observation (see 2.2), thus the spin up is not necessarily too short. We have also found that with a cyclic lateral boundary condition without topography, the modeled results converge with profiles rather quickly. Based on the results of testing runs and actual simulations, we believe that the impact of the spin-up time on simulations is quite limited.

*- Bottom of page 11, other reasons include incorrect surface fluxes, and model weaknesses.*
It has been modified to "The differences between the model and the observation between 13:00 and 16:00 UTC could come from the different representation of simulated result (a domain average) versus that of ceilometer detection (limited to only one vertical direction), the vertical resolution of observed profiles, the limitation of radar in detecting hydrometeors, and in the end, certain model weaknesses likely associated with a lack of hourly radiosondes during the afternoon period to provide sufficient observational constrain".

*- Caption Fig. 7h: why mention SWHR (no lines) at night*
There is a mistake in the discussion of Fig.7h in the text, where SWHR (bottom horizontal axis in the Figure) should be LWHR (top horizontal axis). This has been corrected.

*- Lines 370-371: The increase in cooling with higher N is only true for clouds with LWP < 25 g/m2.*
Solid point, LWPs of many cloud blocks during the stratiform stage meet that criterion (Fig. 5).

*- Line 595: references?*
Since the according discussion with references has been provided in the main text (e.g., Section 4), thus, we would like to limit the citation as a common practice in Summary and Conclusion. We have added words of "as demonstrated in this study and several previous ones" in the revised manuscript.

---

## Referee Report (RR1)

**Review of "The Impact of Aerosols on the Stratiform Clouds over southern West Africa: A Large-Eddy Simulation Study"**

**Summary**

The authors have included more comparisons, relevant technical details, as well as cleaned up confusing statements, improving the manuscript from its previous version; however, I find that some questions were not fully addressed. From my side, some of them were aimed to ensure that the work is reproducible, so I'm including them in minor comments. Regarding the grammar and style of the manuscript, there is still room for improvement. Nevertheless, their work is still highly relevant and the scope of the paper is now more clear.

**Minor comments**

- The work is structured in a first part where the REF case is validated, and a second part where the aerosol experiments are performed. I suggest minimizing the description of the REF case as much as needed. For example, I don't see any use of reporting how much is the temperature at cloud top in every hour of the simulation. The relative differences between model results and observations are key but too much description is overwhelming for the reader, what is really useful from there?

- One of the interesting findings of this work is pointing towards how a polluted scenario can increase spatial cloud variability, which was reported as a research gap in the Introduction. I don't think this is clearly stated in the abstract right now. It motivates it around L30 and then it does not mention spatial variability again. The conclusions summarized it better. Was this process observed/reported in other studies?

- Following this topic, it may be worth exploring how different are the trajectories of the POL, REF, and CLEAN cases in a space defined by cloud fraction and reflectivity. This could support the description around L695.

- Are there observation based reports finding that solar variability is also greater for more polluted cases?

- Some of my previous comments were aimed towards documenting details of the setup so that the work can be reproduced rather than questioning if they were correct. I'm bringing back some of those topics so that you can evaluate if they can be included:

    - What are the initial conditions of this run like?
    - How are the vertical profiles created? Just applying the nearest value to the corresponding grid height? Or is there any averaging performed?
    - I now see that the subsidence velocity is quite high, is this typical of this region? Or was that needed in order to control PBL growth to match observations?
    - The way to find the inversion is specific for this case, right?

- There is still room for improvement in the readability of the document. I suggest asking for professional help or using one of the many tools available for checking grammar and style. Below are included some suggestions but I didn't have time for a more thorough check.

- Fig. 5 is still stretched

- What is this increase of the vertical wind speed referring to in L525?

**Typos / writing suggestions**

- General/style: using more articles like "the" and "a" could help.

- L19 modeled or modelled?

- L35 lower?

- L39 break up

- L86 A large amount

- L120 is denote the correct verb?

- L124 highlighted

- L148-149 redundant mention to semi-direct effects

- L176-178 redundant mention to surface fluxes

- L273 portions

- L280 but they are

- L289 thermodynamic

- L299 layers

- L307 This last part of the sentence is a bit confusing.

- L324 you can include how much it varies: 400 to 1200 m

- L354 main goal

- L397 The spatial behavior of LWP

- L397 cloudy

- L404 Midcloud, CPP decreases...

- L405-406 less inhomogeneous or less homogeneous?

- L409 values

- L413 160:00 UTC.

- L420 add comma after domain

- L421 Locally? LOcally?

- L427 lower than

- L434 "almost" instead of "near"

- L464 observed at Savè

- L471 1°C is not much of a decrease, same for other statements in this section

- L488 maybe it's worth mentioning that these processes are not represented in LES

- L494 later?

- L515 Maintaining a stratus layer...

- L521 reduction

- L525 "and then its increase instead" of "it rises"

- L545 ...24°C while the cloud top temperature is 20°C

- L553 nearly

- L554 Based on the observations, do these upper clouds yield a lower observed solar irradiance?

- L562 ground temperature

- L562 I don't know if this is cloud break-up time, as clouds were indeed already broken then

- L564 I don't understand this well, is it related to the forcing tendencies?

- L580 cloud break-up

- L600 in depth

- L623 Break this sentence into shorter ones

- L639 What does "10 superior" mean?

- L666 decrease

- L695 account for

- L700 "but" instead of "while"

- L713 the inversion layer

- L758 more slowly

- L800 impact

---

## Author Response (AR2)

**Responses to the Remarks of the Handling Editor Dr. Graham Feingold**

Dear Graham,
We very much appreciate your effort in handling our manuscript.

We have carefully addressed each of the new comments from both reviewers. In the meantime, the manuscript has also been revised accordingly, with an emphasis on adding more discussions about relevant previous works, mostly in Section 4. In responding to the recommendation from both reviewers, we have also reduced the content of Section 3 that discusses the results of REF run. Additionally, we have further improved the readability of the paper.

We would also like to respond to your specific note of: "Finally, I do not see any responses to my comments, or an attempt to address them in the manuscript. I reattach those comments here".

Our recollection is that: we received your remarks sometime after our submission of the responses to the two reviewers' previous comments alongside a revised manuscript to the ACP in early April. We then made an extensive effort to carefully address each of your comments and to conduct a massive revision of the manuscript. Specifically, new figures and associated discussions of time variations of LWP and CF in each of the simulations, and the discussions of certain previous works were added. We successfully submitted our responses to your comments to the ACP site on April 25. As ACP has made it available to the public ever since in the interactive discussion stie of our paper, under EC1 pile, we thought that you should have already read it. On the other hand, we could not, however, upload the heavily revised manuscript in responding to your comments (for convenience we noted it as the RE version here), this is due to the ACP uploading rules. Thus, we decided to wait the new comments from the reviewers to merge additional revisions, then upload a combined new version of the manuscript. As a special note, we have now described this matter to both reviewers and informed them that the new revision in responding to their additional comments has been made based on the RE version, which had already addressed many of their new concerns along with yours. We understand the difficulty to track every paper you are handling, and we again appreciate your effort.

Best regards!

**Responses to the Additional Comments of the Reviewers**

We very much appreciate the additional comments from both reviewers. These comments have led to a further improvement of our manuscript.

We would like to specifically indicate that, soon after we submitted the responses to reviewers' first round comments alongside the revised manuscript, we received the comments from the handling editor, Dr. Graham Feingold. Along with the effort in making our responses to the editor's comments, we also significantly revised the manuscript (hereafter referred as the *RE version* for convenience), where many sections were massively rewritten, and numerous typos or grammatical errors corrected. Most importantly, additional figures and associated analyses of e.g., time evolutions of LWP and CF (we used cloud fraction instead of other diagnostics in this version) in different model runs. More discussions of certain previous studies were added as well. While the responses to the editor were successfully submitted in the ACP website and soon became available to the public (under the EC1 pile), the upload of correspondingly revised manuscript, the RE version, however, was not possible due to the procedure of ACP. Therefore, we decided to merge any additional revisions in responding to the reviewers' new comments into the RE version. The modifications made in the RE version can be easily found in the tracked changes from the currently submitted version of the manuscript. In fact, many of the new comments from the two reviewers have already been addressed in the RE version as would be indicated in our following responses.

The following are our point-by-point responses to the reviewers' comments (here the reviewers' comments are displayed first in bold *Italic* font).

1. **Responses to the Additional Comments of Dr. Mónica Zamora Zapata**

*Summary*
*The authors have included more comparisons, relevant technical details, as well as cleaned up confusing statements, improving the manuscript from its previous version; however, I find that some questions were not fully addressed. From my side, some of them were aimed to ensure that the work is reproducible, so I'm including them in minor comments. Regarding the grammar and style of the manuscript, there is still room for improvement. Nevertheless, their work is still highly relevant and the scope of the paper is now more clear.*

We truly appreciate these positive remarks made by Dr. Mónica Zamora Zapata.

*Minor comments*
*• The work is structured in a first part where the REF case is validated, and a second part where the aerosol experiments are performed. I suggest minimizing the description of the REF case as much as needed. For example, I don't see any use of reporting how much is the temperature at cloud top in every hour of the simulation. The relative differences between model results and observations are key but too much description is overwhelming for the reader, what is really useful from there?*
The point is well received. We have shortened the REF analysis and made the discussions focus mostly on comparisons with observations rather than providing a hourly report of, e.g., temperature change.

*• One of the interesting findings of this work is pointing towards how a polluted scenario can increase spatial cloud variability, which was reported as a research gap in the Introduction. I don't think this is clearly stated in the abstract right now. It motivates it around L30 and then it does not mention spatial variability again. The conclusions summarized it better. Was this process observed/reported in other studies?*

Thank you for this excellent point. It has been one of the emphasized enhancements of the RE and the current revision with a specific discussing point of cloud fraction. The Abstract has been rewritten to provide more precise highlights of our findings including the aspect indicated by the reviewer here. In addition to the aerosol concentration factor, the impact from the semi-direct effect by BC has also been highlighted in both Abstract and Conclusion.

Regarding previous works, for example, we have cited Wang et al., 2003 who suggested that cloud under lower aerosol concentration could form thinner cloud layer more easily than in polluted cases, though the model capacity alongside aerosol configuration in that study is differ from ours. We have also discussed Dearden et al. (2018) who conducted a LES simulation of another DACCIWA case though with a passive aerosol configuration, where they run a paired simulations with a binarily configured sedimentation of droplets (i.e., inclusion vs. exclusion). If one simply interprets the sedimentation as a function of droplet size (thus negatively correlated to aerosol concentration), then the derived variation of LWP and CF in that work qualitatively agrees with our finding of LWP and CF changes in responding to aerosol concentration. These are just two examples among those we have added and discussed in the revised manuscript particularly in 4.2 and 4.3.

*• Following this topic, it may be worth exploring how different are the trajectories of the POL, REF, and CLEAN cases in a space defined by cloud fraction and reflectivity. This could support the description around L695.*

This has been greatly enhanced in the RE and the current version using cloud fraction as the discussing point, aided by additional figures showing the time evolution of both LPW and CF, most importantly, as functions of aerosol concentration as well as chemical composition. Besides the revision in this paragraph, an additional paragraph has been added to enhance the discussion.

*• Are there observation based reports finding that solar variability is also greater for more polluted cases?*

We have not been able to identify such reports other than estimates based on satellite retrieval.

*• Some of my previous comments were aimed towards documenting details of the setup so that the work can be reproduced rather than questioning if they were correct. I'm bringing back some of those topics so that you can evaluate if they can be included:*
*– What are the initial conditions of this run like?*

The model was initialized from sounding at 23:00UTC of July 2 as described in the manuscript, with a one-time perturbation to get the turbulent mixing to start. The thermodynamic and dynamical profiles in the early hours are shown in Figure 7, these data can be obtained from DACCIWA website as indicated in the Code and Data Availability section.

Also, as a general response, we would like to indicate that the entire Model and Data description section has been largely rewritten. Several components of the model such as those regarding aerosol microphysics and chemistry have been added in the RE version, along with certain

details of model configurations (including those in Section 4.1 for providing additional information and explanation for the sensitivity simulations).

*– How are the vertical profiles created? Just applying the nearest value to the corresponding grid height? Or is there any averaging performed?*
Directly since the vertical resolution is reasonably high.

*– I now see that the subsidence velocity is quite high, is this typical of this region? Or was that needed in order to control PBL growth to match observations?*
*– The way to find the inversion is specific for this case, right?*
Answers to both are yes, these can be done rather straightforwardly based on the soundings.

**• There is still room for improvement in the readability of the document. I suggest asking for professional help or using one of the many tools available for checking grammar and style. Below are included some suggestions but I didn't have time for a more thorough check.**
One of our "senior" co-authors has made an extensive effort to polish the readability.

**• Fig. 5 is still stretched**
We have resized the figure.

**• What is this increase of the vertical wind speed referring to in L525?**
The sentence has been modified to "…, implying an increase of surface solar heating".

*Typos / writing suggestions*
We have generally adopted the reviewer's suggestion in performing the according revision hereafter. Therefore, here we only list the responses where additional answers would need.

**• *General/style: using more articles like "the" and "a" could help.***
Done with a best effort.

**• *L35 lower?***
Here lower is used as a verb.

**• *L324 you can include how much it varies: 400 to 1200 m***
We have revised the sentence to:
"However, the nocturnal-diurnal life cycle in our case involves a dynamically evolving cloud top from 400 to 1200 m, particularly in the daytime, making it a difficult task to prescribe a highlighted zone for finer resolution. Our fast-testing results, on the other hand, did not suggest an alarming difference between the run with 10 m and 5 m vertical resolution (not shown)".

**• *L404 Midcloud, CPP decreases...***
**• *L405-406 less inhomogeneous or less homogeneous?***
We have modified the sentence to:
"Between 10:00 and 13:00 UTC, CF of the layers between domain mean CBH and CTH starts to decrease from near 100% to 90%, while CF at CBH and CTH decreases more substantially to reach near 60% and 80%, respectively. This leads to a less homogeneous cloud deck…".

**• L554 Based on the observations, do these upper clouds yield a lower observed solar irradiance?**
We do not have direct observational evidence for this.

**• L562 I don't know if this is cloud break-up time, as clouds were indeed already broken then**
Yes, stratus broke up while separate convective cloud blocks still existed.

**• L564 I don't understand this well, is it related to the forcing tendencies?**
Primarily surface solar radiation.

All the other issues have been resolved either in the RE version already or this the current revision by adopting the reviewer's suggestions.

**2. Responses to the Additional Comments of the Reviewer #2**

*I find that some of the major and minor issues I raised have not been fully addressed.*
*Major issue #1:*
*-Place results in context with previous studies, both when describing the reference case as well as the aerosol sensitivity studies.*

*I appreciate that a new paragraph has been added in the beginning of Section 4. However, I am afraid such paragraph is too general and does not provide any relevant information into how the presented study relates, or differs, from those mentioned by the author therein. I would appreciate more references and comparisons to other studies both in Section 3 and 4. In general, I find that there is discussion missing in this article. It could be done along the result description, as I suggested first, or in a separate discussion section.*

Additional discussions on the similarity and/or differences between our results and those of others have been added in the revised manuscript. Please also note that we have significantly revised the Introduction to highlight the major difference between our modeled case and most others (e.g., land vs. ocean), along with certain different controlling factors as well as expected feedbacks (e.g., surface heat flux responses etc.). New discussions in Sector 4 have been added particularly regarding the aerosol impacts in a context of different aerosol model configuration and profiles.

*Major issue #2:*
*-Language-related issues, like grammatical errors and long and complex sentences make following the manuscript a difficult task. Please correct all grammatical mistakes (only some examples given along the line by line comments) and keep sentences short for the shake of clarity.*

*I can see that a number of errors, misspellings and confusing sentences have been reworded. I think enough has been done in this respect.*

We appreciate the positive remarks made by the reviewer.

*Minnor issue #2:*
*-Section 3 goes through the results of the REF simulation in high detail. Readers would find it*
*easier to understand, however, if instead of a description of each result, a more concise section*
*with the most relevant results is presented. This would also allow, as suggested in the fist*
*Major issue, some room to link the relevant results to previous studies.*

*I do not think much action has been taken in this respect. I agree that, to a certain extent, this*
*is matter of a style. However, I believe that the current text makes it challenging for a not very*
*expert reader to understand what the new findings are, and what is common knowledge from*
*previous research from such a detailed and indiscriminate description of the results.*

The same point from both reviewers is well received. As a response, we have greatly reduced the discussions in REF section, particularly of 3.2. In addition, we have rewritten the summary paragraph in the end of 3.1 to provide a highlight of the overall comparison of our modeled results with observations. In the meanwhile, we decide to keep certain contents of 3.2 to benefit potential reproduction works in near future, besides some later analyses in the paper on the model-observation inconsistency. Here are three examples among others of the newly added discussions in Section 4.2 and 4.3 (note also the new figures shown time variations of LWP and CF):

"The difference between CLEAN and REF in cloud macrophysical features such as CBH and CTH is visible though largely limited to a few tens of meters. However, their differences in cloud fraction and microphysical features are rather significant. As expected, from formation to break-up of the clouds, $N_c^{CLEAN}$ is lower than $N_c^{REF}$ and $r_c^{CLEAN}$ is larger than $r_c^{REF}$. At 02:00 UTC, $N_c^{CLEAN}$ has a maximum value of 181 *droplets cm$^{-3}$* and $r_c^{CLEAN}$ of 7.58 *µm,* in comparison to 653 *droplets cm$^{-3}$* and 5.1 *µm* for $N_c^{REF}$ and $r_c^{REF}$ respectively with the same liquid water content value (0.35 *g m$^{-3}$*). $r_c^{CLEAN}$ further increases to 12.55 *µm* at 08:00 UTC, then decreases slowly to a maximum value of 10.97 *µm* at 14:00 UTC with $LWC^{CLEAN}$ reaches near 0.45 *g m$^{-3}$* instead of 0.49 *g m$^{-3}$* for $LWC^{REF}$, likely due to an increased activation ratio of aerosols after sunrise. Despite a relatively larger droplet size in CLEAN than POL and REF case, there is no clear sign of massive formation of drizzles even during the convection stage (Fig. 10). Nevertheless, sedimentation thus evaporation of larger droplets from entrainment zone and cloud base could likely create a thermodynamic perturbation (*e.g.*, Stevens *et al*., 1998; Jiang *et al*., 2002). In a LES simulations using passive aerosol profile for July 4-5 DACCIWA case, Dearden *et al.* (2018) found that the sedimentation would remove droplets from the entrainment zone thus, through a feedback, lead to a cloud deck with higher LWP while smaller CF than the case where sedimentation is completely excluded. This could imply a similar contrast between CLEAN and the two polluted cases in our simulations, by simply assuming the total sedimentation amount is proportional to the droplet size (*i.e.*, inversely to the CDNC), though the quantity of such a perturbation seems rather small here, not to mention the more sophisticated feedback involved in our case introduced by the dynamic aerosol-cloud interaction in our model".

"Looking into various timely varying metrics of LWP in different model runs, we find that in general, LWP is inversely promotional to CDNC, as LWP in POL < LWP in REF < LWP in CLEAN, and this is applied to different metrics of LWP (Fig. 12, Table 3). However, in comparison, the peak LWP varies less significantly in CLEAN case, while peak LWPs in two

other runs decrease with domain averaged quantities in convection stage. There were different opinions regarding the mechanisms behind such an inverse relation between LWP and CDNC (*e.g.*, Ackerman *et al*., 2004; Bretherton *et al*., 2007), not to mention that most such hypotheses were proposed based on the cases of marine low clouds that might not be directly applied to the cases over land. In our analysis, the difference in turbulent mixing driven by the surface radiative heating, as influenced by different microphysical features in various cases, seems having played a critical role. The situation of cloud fraction (CF) is somewhat more complicated. As shown in Table 3 and Fig. A4, CF relation with CDNC varies in different stages. An inverse relation between CF and CDNC generally stands in the earlier and later period of the convection stage, in the middle of the convection stage (13:00-15:00 UTC), the above relation, however, would reverse, alongside the vertical cloud extent as discussed previously".

"The above results have demonstrated the important role of solar absorption by aerosols in determining the life cycle of LLSCs. The atmospheric heating by light absorbing BC would limit the elevation of cloud top, especially during the break-up stage (Koch and Del Genio, 2010b; Zhang and Zuidema, 2019). Such a heating can also increase cloud fraction then delay break-up until late afternoon, especially for clouds with higher cloud droplet number concentration in polluted environment such as in POL and REF runs (opposite to the outcome by considering aerosol number concentration only), and thus affect the indirect effect of aerosols. Note that our modeling configurations are based on the aerosol profiles that are relatively well-mixed throughout the PBL then with concentration gradually decreasing along altitude above PBL. Certain previous sensitivity experiments suggested that the location of BC layer within or above PBL could have different impacts on the development of convection, entrainment, and thus life cycle of the low clouds within PBL. For instance, Johnson *et al*. (2004) suggested that without considering the indirect effect of aerosols, BC existing within boundary layer would lower LWP by nearly 20% in a marine low stratocumulus case, where the cloud response is less sensitive to the surface shortwave heating change comparing to the situation in our case. Feingold *et al*. (2005) found that smoke plumes containing BC near the surface would reduce the cloudiness through both the atmospheric heating and weakening effect on surface heat fluxes by BC. These results though obtained with somewhat different model configurations than ours (e.g., coarser vertical resolution, different surface, etc.) are in a qualitative agreement with our findings. Nevertheless, the unique configuration of our model allows us to quantitatively examine the semi-direct effect with varying aerosol chemical compositions and thus extent of aerosol absorption. This has led us to reveal further insights of the complicated interplays among various aerosol effects besides their individual impacts on the life cycle of LLSCs".

---

## Author Response (AR3)

Response to the Editor's Comments
(Here the editor's comments are displayed with Italic and bold font)

*Dear Colleagues,*
*I have taken a close look at the revised manuscript. I am still very concerned about the readability of the manuscript, which is compounded by the significant grammatical and typographical errors. I have consulted with the editorial staff on this matter. While I do understand that ACP will perform a technical edit on this document prior to publication, I am particularly concerned that most readers will be frequently confused by the \*meaning of the text\*, and that this will obstruct the exchange of information and the transfer of physical insights. This will not serve the authors well.*

*My comments below are an attempt to highlight some of the parts of this document that need particular attention. There are likely others that I have not picked up on. Please also note that while I marked these as 'minor revisions', there are some questions that are much more substantive. Until these very important corrections are made, I will not be able to accept the manuscript for publication in ACP.*

*Graham Feingold*

*Note 1: Page and line numbers refer to the diff file.*
*Note 2: I have not attempted to correct the many grammatical errors that unfortunately make the text very hard to follow at times.*

We thank your effort in helping us to improve the manuscript. The following are our point-to-point responses to your comments. Certainly, our revisions go beyond these indicated modifications as shown in the manuscript version tracking the revisions. Specifically, the section 4.1 has been merged with 4.2, and 4.3 is now 4.2, both sections, alongside many other parts of the manuscript, have also been largely rewritten to improve the readability as well as to interpret the results more adequately. Several redundant or inadequate paragraphs and statements have also been removed.

*Page 1, line 38. Please be clear: remove the word 'can' and make it clear exactly what the model shows.*
Done.

*Page 2 the first paragraph at the top of page 2 is not clear at all to me. There also are grammatical errors that make it difficult for any reader to follow.*
We have removed the sentences containing "positive" and "negative contribution" here (and in other places such as Conclusion) to make the paragraph read better, and to reflect what the model shows as suggested in the above comment. It reads now as: "In addition, we find that an excessive atmospheric heating up to 12 $K\ day^{-1}$ produced by absorbing black carbon aerosols (BC) in our modeled cases lowers the height of cloud top and liquid water path, resulting a weaker extent in vertical development while a higher cloud fraction and delaying intense cloud break-up before later afternoon. While the thinner clouds resulted from such a heating, on the other hand, would break up faster in late afternoon when convection is further strengthened."

*Page 5, line 139 please also refer to Bretherton et al. 2007 GRL, who were the first to identify the impact of cloud droplet sedimentation.*
This paragraph is about certain previous modeling efforts using DACCIWA data (now opens with a brief statement to make this more clearly). Nevertheless, your point is received, we have modified the sentence to: "The impact of sedimentation on LLSCs was indicated by previous studies (e.g., Bretherton et al., 2007). This issue has also been addressed in a modeling of DACCIWA case by…".

*Page 8, line 257. You discuss evaporation of this light precipitation. Please expand on the physical chain of events -- otherwise the reader does not gain any physical insights.*
It has been modified to: "…by the cooling alongside moistening brought by the evaporation of this light precipitation, which could enhance the liquid water path of the beneath LLSC…".

*Page 9, line 297: you mention a pseudo prognostic approach for supersaturation was developed. What does this mean? For example, the last line at the end of this page discusses the Abdul-Razzak and Ghan parameterization, which is a diagnostic supersaturation calculation. Therefore, I am quite confused about what you mean above, and how exactly S is calculated.*
It has been modified as: "…a pseudo-prognostic approach correcting the diagnostically derived…".

*Line 317 on page 9: 'secondary' organic aerosol not 'second' organic aerosol.*
Done.

*Page 10. Line 328 – 'module' not 'nodule'.*
Done.

*Page 10 line 345 you say that the liquid cloud affective radius is computed from the liquid water content. Please be clear that the drop size is equally determined by the drop concentration.*
Added "and droplet number concentration".

*Page 10 line 365: you speak about an absorbing layer, but what does the layer absorb? Please be precise in your language.*
It has been modified to: "a "sponge layer" is set between 1.8 and 2 km height to absorb wave reflection".

*Page 11 online 382 you mentioned an 'alarming difference' please quantify what you mean by 'alarming'. This is hyperbole.*
Changed to "…any significant…".

*Page 12, lines 414 to 424: these lines repeat text that was mentioned on lines 262 to 270. However, they are inconsistent with the previous text. Please get the numbers straightened out.*
The numbers in 414-424 are corrected values of the original measurements (262-271), this was

indicated in Line 271. Nevertheless, the sentence in 415 has been modified to: "derived from the corrected original measurements as described by…".

*Page 13. Line 446 you mentioned 'limited to only one vertical direction'. Please change this text. I think you mean 'limited only to the vertical direction'.*
Thanks, done.

*Page 13, figure: there is a red line at roughly 600 m. This line should not cross the color bar.*
This is the tracking version of the manuscript. The figure in P.13 was deleted, that's why the red line was there to mark its removal. The new figure was displayed in the next page.

*Page 18. Please mention that the timing of the surface flexes is incorrect. In other words, there is a temporal offset in the surface flux relative to the observations.*
We assume this was referring to the surface heat fluxes. We have modified the corresponding sentence to: "Between 09:00 and 14:00 UTC, the modeled sensible and latent heat fluxes follow the measured trends though with a clear temporal offset, leading to an overestimate…".

*Line 561: you mention supersaturation in an updraft at cloud base. However, supersaturation is based on the Abdu-Razzak and Ghan parameterization so this is not necessarily cloud base supersaturation.*
"at cloud base" has been removed.

*Page 21, line 573: you mention that the drop concentration reaches 1750 per cc, however figure 7e shows a much smaller concentration of drops.*
Thank you! It has been modified to "above 1200" as shown in the figure.

*Page 21, line 574 you mention that this high drop concentration is most likely due to the continuous activation of aerosol into cloud droplets. However, you do not discuss the fact that such high drop concentrations would suppress the supersaturation and therefore significantly limit further activation, unless there are significant vertical accelerations. I'm concerned about whether these calculations are correct. Are aerosol particles removed locally from the population once they have been activated? Are they returned after a drop evaporates?*
The model does track aerosol population in and out of condensed waters and thus transfers it between the two states accordingly when activation or evaporation occurs. This sentence mostly repeats the previous one. Also, the numbers cited here are domain averaged values. For the first 2 hours of the cloud formation, the explanation is apparently not complete. Therefore, it has been removed.

*Page 21 line 585 you discuss the influence of drop concentration on longwave cooling in figure 7h. Please make it clear that the sensitivity only occurs at low liquid water path, and please quote the relevant literature.*
The sentence has been modified to: "For LLSCs at this stage with many low LWP blocks, the more numerous the cloud droplets are the stronger the cooling is (e.g., Petters et al., 2012), …".

*Line 595: what do you mean by 'proper' temperature and humidity conditions?*
It has been changed to "…needs stable ground temperature and moisture supply".

*Page 22, line 600 what do you mean by 'settled'?*
Removed.

*Page 23 line 624 please correct the units to watts per meter squared.*
Done.

*Line 642. Please remove the word 'intense'. This is hyperbole. Or alternatively please quantify the surface heating.*
Done.

*Page 24 line 666, same comment: please remove the word 'drastically'.*
Done.

*Page 24, lines 683 the word 'apparently' is not clear. Do they or don't they?*
Removed.

*Page 27 line 772: you mentioned 'massive formation of drizzles' what does massive formation of drizzles mean? Please remove the word massive and quantify as necessary. Perhaps you mean 'significant drizzle'? Quantification puts things in perspective.*
Changed to "significant drizzle".

*Line 776 again I believe you should quote Bretherton et al., 2007.*
Added "Consistent with certain previous findings (e.g., Bretherton et al., 2007), in a …".

*Line 779. You discuss total sedimentation amount. Does your model simulate cloud droplet sedimentation? Most bulk models do not. This is really important because if you are discussing this mechanism of cloud droplet sedimentation, then it is important that you connect the process to the physics in your model.*
We have modified "total sedimentation" to "total drizzle sedimentation", also, we have made it more clearly in the model description (2.3) that the KHKO scheme differentiates cloud droplets from drizzle drops using 25 micron radius as the boundary, the sedimentation of the latter is calculated.

*Page 39, 831 please change the word 'timely' to 'temporally'.*
Done.

*Line 832. Please change the word promotional to proportional.*
Done.

*Line 840 to 845: please give some insight as to why. Can you advance knowledge?*
The sentence has been modified to: "The situation of cloud fraction (CF) is somewhat more complicated. As shown in Table 3 and Fig. A3, CF relation with CDNC varies in different stages. An inverse relation between CF and CDNC generally stands in the earlier and later period of the convection stage. This is primarily due to the faster evaporation of clouds with higher CDNC driven by entrainment in the former period (note the controlling role of CF in determining the

surface incoming solar radiation and thus turbulence in this stage), or by strong convection in the latter. In the middle of the convection stage (13:00-15:00 UTC), the above relation, however, would reverse or become insignificant, owing to a weaker turbulent mixing in polluted cases since the cloud reflectivity becomes the dominant factor in controlling the surface incoming solar radiation as discussed previously. Therefore, an analysis throughout the entire LLSC life cycle is very important to understand the response of CF alongside LWP to aerosol variation. Note that the atmospheric heating caused by absorbing black carbon aerosol is already included in this series of sensitivity simulations, though its impacts on the above result will be discussed later based on another set of sensitivity runs."

*Page 31 line 849. The text reads 'when dynamical situation is more complicated to maintain a constant liquid water content'. I do not understand what you mean by this and I suspect many readers will not understand either.*
Removed.

*Line 855. The text reads 'Our study weights in both size, distribution and chemical composition'. This text is not understandable to me.*
It has been modified to: "Our sensitivity simulations utilize different aerosol profiles that reflect the variations in both aerosol concentration and chemical composition based on observations, …".

*Page 32. The caption for figure 12 has undefined acronyms AODON and AODOFF these are not defined. I suspect these are typographical errors.*
Done.

*Figure 12 is described in the text on Pg 30 as a comparison between the three different cases: polluted, clean, REFF. So why have you included the ADEON and ADEOFF cases in this figure?*
*If you want the reader to refer to REFF, polluted, clean including the aerosol coupling (ADEON), then you should provide a separate figure. And you seem to hardly compare the others. I got so confused that I gave up.*
Indeed, the figure was used in both 4.2 and 4.3 (now 4.1 and 4.2), partly for the purpose to reduce the number of figures. We have added "Fig. 12, ref. ADEON curves" in the discussions in 4.2 (now 4.1). The discussions comparing ADEON and ADEOFF are in new 4.3, e.g., Line 914-918, among other places.

---

## Author Response (AR4)

**Responses to the comments of the editor**
(The comments are displayed with Italic and bold font)

*1) Lines 32-37: I don't understand the message. This is the abstract and must be crystal clear and in correct English.*
It has been revised to: "In addition, by comparing the simulations including versus excluding an excessive atmospheric heating up to 12 $K\ day^{-1}$ produced by absorbing black carbon aerosols, we find that the semi-direct effect resulted from this heating is to lower the cloud top as well as liquid water path while increase cloud fraction, thus to delay the intense cloud break-up until the later afternoon when convection is further strengthened.".

*2) Pg 2, lines 40-46 are understandable but I am concerned that the technical editors might correct the grammar and change what you intend to say.*
We have revised the sentences to: "Low-level stratiform clouds (LLSCs) have a higher albedo and a larger cloud cover than many other types of clouds (Hartmann *et al*., 1992; Chen *et al*., 2000; Eastman and Warren, 2014). Their reflection of solar radiation is important to Earth's radiative budget.  LLSCs often occupy the upper few hundred meters in the planetary boundary layer (PBL). Their appearance can be persistent when associated with a high-pressure system with a large-scale subsidence that stabilizes the PBL. LLSCs are often formed over cooler subtropical and mid-latitude oceans, constantly covering more than 50% of these areas (Wood, 2012)".

*3) Lines 71-72: unclear*
It has been revised to: "Indeed, it is still difficult to estimate the indirect effect of aerosols and thus to minimize the uncertainty associated with this effect in the climate models (Boucher *et al*., 2013; Li *et al*., 2022)".

*4) Lines 74-77: unclear*
Revised to: "Previous studies had investigated aerosol-cloud interactions in LLSCs using high-resolution Large-Eddy Simulation (LES) models. Many of these studies were on the cases over ocean (*e.g*., Ackerman *et al*., 2004; Sandu *et al*., 2008; Twohy *et al*., 2013; Flossmann and Wobrock, 2019), where latent heat flux at the surface plays a more important role in the life cycle of LLSCs than sensible heat, while the latter dominates in the cases over land (Wood, 2012; Ghonima *et al*., 2014)".

*5) Lines 129-130: please make this understandable.*
Changed to: "Majority of them, however, only addressed the response of LLSCs to the aerosol abundance rather than aerosol effects associated with different chemical compositions (such as the semi-direct effect of black carbon) by taking the advantage of measurement data obtained during the field campaign".

*6) Line 240: you never mentioned the boundary conditions for the model. Is it doubly periodic? How do you apply forcings?*
These are described in the Model settings section. We could rearrange the subsections.  While we do not believe this would be a critical issue since the reader can have the information a few paragraphs later. We thus decide to keep the current structure.

*7) Line 257: "You should say that LIMA predicts mass as well as number.*
Added "mass as well as".

*8) Line 263: "Only the sedimentation of drizzle…"*
Done.

*9) Line 280: You still haven't explained which S you use. If it's A-R & Ghan then it's not a pseudo prognostic approach. Which one is it??*
Abdul-Razzak and Ghan is used in ORILAM. We have indicated this by adding "(though without the correction of Thouron et al., 2012)".

*10) Line 298, please tell the reader what value of k (inverse of distribution width) that you used.*
For continental condition since airmasses arrived Save came from various continental sources as suggested by the measured aerosol properties. Indicated in the revised text.

*11) Line 305: How many subtiles did you use?*
It is described two paragraphs down in the Model settings, as "…using data from Savè supersite, with the typical vegetation consisting of shrubs, crops, or taller trees, assuming a flat surface in the area around Savè".

*12) Line 312: "alongside" ◊ and*
Done.

*13) Line 410: "a more homogeneous"*
Done.

*14) Line 458 "reasonably well", not "successful"*
Done.

*15) Line 507: Why "blocks"? Don't you mean columns? Also, you must mention that the low LWP columns are < 25 g/m2 or this makes no sense at all.*
Now reads as: "…columns (*e.g.*, with LWP < 25 g/m$^2$; Petters *et al*., 2012),".

*16) Lines 635-641: This shows that the authors do not understand the issues. Your model doesn't represent sedimentation of cloud drops. The sedimentation of large drops is not responsible for this process. Therefore, the sedimentation feedback (Bretherton) does not apply. Perhaps the evaporation entrainment feedback by Wang, Wang, and Feingold (JAS, Vol 60,2003) is acting. But regardless, these lines of text are wrong and should be removed or rewritten after a careful read of the relevant papers.*
These statements (with modification) perhaps are still useful here to indicate the additional feedback of LLSC proposed by previous studies. We have modified the section to make it more clearly as: "Note that sedimentation thus evaporation of larger droplets (smaller than drizzles though) from entrainment zone and cloud base could likely create a thermodynamic perturbation as well (*e.g.*, Stevens *et al*., 1998; Jiang *et al*., 2002). Consistent with certain previous findings

(*e.g.*, Bretherton *et al.*, 2007), in a LES simulations using passive aerosol profile for July 4-5 DACCIWA case, Dearden *et al.* (2018) found that the sedimentation would remove larger cloud droplets from the entrainment zone thus, through a feedback, lead to a cloud deck with higher LWP while smaller CF than the case where such a sedimentation is completely excluded. Since the sedimentation of non-drizzle droplets is not included in our model, our results cannot be used to directly address this issue".

*17) Line 675: remove "much"*
Done.

*18) Line 685-688: You say "Interestingly". Please explain why!*
Modified to: "Modeled clouds in POL and REF appear to dissipate earlier and much faster than in CLEAN in the later afternoon, largely due to their smaller droplet sizes (Fig. 11, bottom panel)".

*19) Lines 697-700: Again, you're not representing the sedimentation of small cloud drops so this does not apply. So, the "different opinions" are not relevant here. Again, these lines of text are wrong and should be removed or rewritten after a careful read of the relevant papers. Maybe the Wang et al. (2003) mechanism is relevant but you will need to be the judge of that.*
Removed.

*20) Line 730: I don't think your references are appropriate given that Bretherton showed that the Ackerman result was a sedimentation-entrainment feedback. The best you can say is that the entrainment of dry air evaporates cloud water more efficiently when the drops are small (Wang et al. 2003).*
Ackerman et al., 2004 is now replaced by Wang et al. 2003.

*21) Lines 751-753: "more water vapor tends to condense..". Where do I see this?*
This Is a misquotation and has been corrected. The new sentence now indicates the TKE reduction as shown in the figure.

*22) Line 760: "appear to be" ◊ are (be clear!)*
Done.

*23) Line 761: I cannot see where TKE is reduced below the heating layer.*
It should be "just below the top of the heating layer", and has been corrected now. This can be seen from the panels of 14UTC.

*24) Line 770" "massive" ◊ strong*
Done.

*25) Figure 13: please align the timescale in panel a) with those in b) and c)*
Done.

*26) Fig. 14: y-axis label in 1) and b): are these really differences?*

Yes. As a comparison, the maximum cloudy sky quantity is about 300 W/m$^2$ as shown in Fig. 6(a). Since the discussion of Fig.14 is mainly on the result displayed in the panel (c) while panel (a) and (b) were barely mentioned, we decide to remove the panel (a) and (b) from the figure.

*27) Line 834: You say "Interestingly" but this is not new, so please remove this word, or provide the interesting aspect.*
Removed.

*28) Line 840: You say that droplets are found to evaporate more easily but you didn't show it, you just surmise, since you haven't actually calculated the water budget and the evaporative contribution. Please reword.*
Modified to: "Clouds influenced by higher aerosol concentrations and thus having higher number concentration and smaller sizes of cloud droplets evaporate more easily and this can lead to a lower cloud fraction".

---

## Author Response (AR5)

**Editor's Comment**
Dear Authors,
I hereby accept your manuscript for publication in ACP.
Please upload a revised file with a rewording of the following lines in the abstract for clarity:
"In addition, by comparing the simulations including versus excluding atmospheric heating of up to 12 K day−1 produced by absorbing black carbon aerosols (BC), we find that the semi-direct effect resulting from this heating lowers the cloud top and reduces liquid water path, while increasing cloud fraction, and thus delays significant cloud break-up until the late afternoon when convection is further strengthened".
Sincerely,
Graham Feingold

**Response**
It has been modified to "In addition, our sensitivity runs including versus excluding aerosol direct radiative effects have also demonstrated the impacts specifically of solar absorption by black carbon on the cloud life cycle. The semi-direct effect resulted from an excessive atmospheric heating of up to 12 $K\ day^{-1}$ by black carbon in our modeled cases is found to lower the cloud top as well as liquid water path, reduce surface incoming solar radiation and dry entrainment, and increase cloud fraction".